# Controlling crystallization in covalent organic frameworks to facilitate photocatalytic hydrogen production

Zheng Lin[1,5], Xiangkun Yu[2,5], Zijian Zhao[3], Ning Ding[1], Changchun Wang [1], Ke Hu [3,4] ✉, Youliang Zhu [2] ✉ & Jia Guo [1] ✉

The catalytic performance, depending on the surface nature, is ubiquitous in photocatalysis. However, surface engineering for organic photocatalysts through structural modulation has long been neglected. Here, we propose a zone crystallization strategy for covalent organic frameworks (COFs) that enhances surface ordering through regulator-induced amorphous-to-crystalline transformation. Dynamic simulations show that attaching mono-functional regulators to the surface of spherical amorphous precursor improves surface dynamic reversibility, increasing crystallinity from the inside out. The resulting COF microspheres display surface-enhanced crystallinity and uniform spherical morphology. The visible photocatalytic hydrogen evolution rate reaches 126 mmol $g^{-1}$ $h^{-1}$ for the simplest β-ketoenamine-linked COF and 350 mmol $g_{COF}^{-1}$ $h^{-1}$ for SiO$_2$@COF with minimal Pt cocatalysts. Mechanism studies indicate that surface crystalline domains build the surface electrical fields to accumulate photogenerated electrons and diminish electron transfer barriers between the COF and Pt interface. This work bridges the gap between microscopic molecules and macroscopic properties, allowing tailored design of crystalline organic photocatalysts.

Hydrogen is not only a promising energy carrier with high energy density but also an eco-friendly fuel that merely affords water as a by-product. Artificial photosynthesis of hydrogen from water splitting is one of the most desirable routes for solar energy conversion[1]. During the process, exciton separation, photogenerated charge transfer to surface sites, and surface catalytic reaction toward hydrogen production are all essential steps. The surface properties of photocatalysts are crucial to accumulating photogenerated electrons for dominating proton reduction[2,3]. As vastly reported, inorganic photocatalysts such as metal oxides, metal nitrides, metal sulfides, and bismuth-based compounds have been regulated by various surface engineering strategies including cocatalyst loading[4,5], surface morphology control[6,7], surface modification[8,9], and surface phase junctions[10,11]. In contrast,

polymer organic photocatalysts severely lack studies on surface engineering. Instead, most attempt to rationalize the link of molecular design to photophysical characteristics such as electronic band structure, light absorptivity, and push-pull electronic effect[12–15]. However, little is known about the significance of surface structure control for particulate organic photocatalysts.

Covalent Organic Frameworks (COFs) are a class of porous crystal-line materials consisting of repetitive building blocks connected covalently[16]. COFs feature a series of distinguished properties, including high surface area, adjustable electronic properties, chemically accessible functionalization, and high stability, which make them attractive for photocatalytic hydrogen generation. Over a decade, striking progress has been achieved in promoting the photocatalytic activity of COFs for

[1]State Key Laboratory of Molecular Engineering of Polymers, Department of Macromolecular Science, Fudan University, Shanghai, China. [2]State Key Laboratory of Supramolecular Structure and Materials, College of Chemistry, Jilin University, Changchun, China. [3]Department of Chemistry, Fudan University, Shanghai, China. [4]School of Chemical Science and Engineering, Tongji University, Shanghai, China. [5]These authors contributed equally: Zheng Lin, Xiangkun Yu. ✉e-mail: khu5@alumni.jh.edu; youliangzhu@jlu.edu.cn; guojia@fudan.edu.cn

**a Bottom-up crystallization**

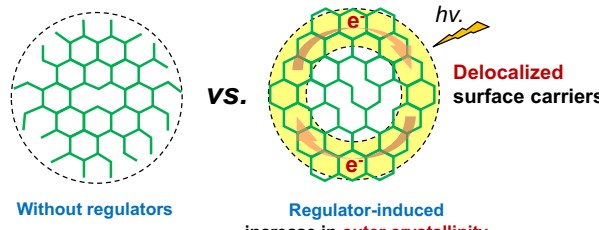

Without regulators

Regulator-induced
increase in **inner crystallinity**

**b Amorphous-to-crystalline transformation**

Without regulators

Regulator-induced
increase in **outer crystallinity**

**Fig. 1 | The effect of growing pathways on zone crystallization of COF.**
**a** Regulator-induced bottom-up growth of COF with outer-to-inner increase in
crystallinity only able to generate localized surface photocarriers. **b** Regulator-
induced amorphous-to-crystalline transformation of COF with an inner-to-outer
increase in crystallinity, conducive to generating delocalized surface photocarriers.

hydrogen evolution[17]. The prevailing strategy is the design of donor-
acceptor systems for COF photocatalysts, aiming at the improved
separation and migration of photogenerated excitons to the surface sites
for proton reduction[14,18]. Also, integrating electron transfer mediators[19],
hydrophilic units[20], photosensitizer components[21], or strong electron-
withdrawing groups[22] into the backbones or side chains endows the COF
photocatalysts with improved hydrophilicity and electronic properties.
However, there has existed a gap between molecular engineering and
photocatalytic performances, as the roles of surface structures deter-
mining the multistep water splitting reaction have long been under-
estimated. For the polycrystalline COF solids synthesized via a bottom-
up route, the surface crystallinity is relatively weaker than the inner
moiety, with numerous remaining defects and unreactive terminals due
to incomplete conversion[23–25]. Minimizing the bulk COFs into nanosheets
or nanospheres enables exposure of more crystalline domains[26–30],
whereas deliberately ordering the whole surface structure is far from
satisfactory yet. It follows that the surface diffusion of photogenerated
carriers is severely confined in the local regions, increasing the possibility
of exciton recombination. Well-organized architecture from surface
building blocks is highly anticipated to tackle this challenge. Enhancing
the surface crystalline domains is conducive to strengthen the built-in
electrical field of surface zones and facilitate interaction with deposited
cocatalysts via π-metal coupling[31,32]. However, to the best of our knowl-
edge, a variety of methodologies available for bulk crystallization have
yet remained invalid for surface ordering arrangement.

Herein, we address a regulator-induced amorphous-to-crystalline
transformation method to intensify the surface crystallization of sphe-
rical COFs (SCOFs) for optimizing surface electronic properties. In our
early report, we demonstrated the controllability of the morphology, size,
and structure of COFs at the microscale via an amorphous-to-crystalline
transformation[33,34]. This pathway also holds great promise for surface
crystallization engineering with regulators, which have often been
incorporated for enhanced crystallinity of bulk COFs[35,36]. The process
involves the first-step synthesis of a spherical amorphous precursor by a
reflux-precipitation polymerization via the Schiff-base reaction. The
obtained precursor features surface-immobilized regulators, given par-
ticle size and uniform spherical morphology. Then, relying on the
reversible aldimine reaction, the amorphous precursor transforms into

the crystalline SCOF by the regulator-mediated local structural rearran-
gement in the typical solvothermal conditions. Such a solid-to-solid
conversion is beneficial to surface ordering as the surface-immobilized
regulators intensify the crystallization kinetics of peripheral moieties,
thereby offering the possibility of an increase in crystallinity of photo-
catalysts' surfaces as opposed to those synthesized via a bottom-up route
(Fig. 1). It is inaccessible to the free mobile regulators as they exert an
influence on the whole bulk phase instead of being confined at the liquid-
solid interface. In addition, the uniform morphology and particle sizes
remain and the coalescence of particulate SCOFs is suppressed as the
reactive sites on the surface are protected with the regulators. In the
photocatalysis test, the regulated SCOF exhibits exceptional photo-
catalytic performance compared to other controls, reaching hydrogen
evolution rates of 126 mmol g$^{-1}$ h$^{-1}$ for the SCOFs and 350 mmol g$_{COF}^{-1}$ h$^{-1}$
for the SiO$_2$-supporting SCOFs. The elaborate study on the mechanism
discloses that surface crystallinity enhancement can effectively prolong
the photogenerated charge lifetime and decrease electron transfer bar-
riers at the interface between the COF's peripheral moieties and Pt
cocatalysts (Fig. 1). Without sophisticated molecular design, the surface-
engineering approach presented here affords a potent means of elevating
the photocatalytic activity of organic photocatalysts.

## Results
### Dynamics simulation of regulator-induced amorphous-to-crystalline growth
To insightfully unravel the regulator-induced zone crystallization
mechanism, we commence the study on the amorphous-to-crystalline
growth dynamics with coarse-grained molecular dynamics (CGMD)
simulation. As displayed in Fig. 2a, b, the hexagonal reticular skeleton
is established by a $[C_3 + C_3]$ CG model, with the same motif topology of
the $[C_3 + C_2]$ model[37]. The two pathways, including one-step bottom-up
crystallization and two-step amorphous-to-crystalline transformation,
were adopted to investigate the surface crystallization dynamics (see
the method details in the Supplementary Information). Figure 2c
exhibits the simulation snapshots for the amorphous and crystalline
spherical models with various radiuses, and Fig. 2d illustrates the bulk
phase crystallization through the one-step route (marked with a star)
and the amorphous-to-crystalline two-step route (marked with a
hexagon), respectively. To reflect the crucial role of regulators in
crystallization, a dynamic bond model[38,39] was employed to describe
the reversibility change in simulations[37,40,41]. The reaction rate con-
stants for the forward and reverse reactions can be expressed
according to the Arrhenius theory, respectively, as

$$K_f = A_f \exp\left(-\frac{E_{bar}}{k_B T}\right), \tag{1}$$

$$K_b = A_b \exp\left[\frac{-\left(E_{bar} + E_{bind}\right)}{k_B T}\right], \tag{2}$$

where T is the temperature in the unit of Kelvin, and $k_B$ is the Boltz-
mann constant. As reported in our early work, $E_{bar}$ is approximate to
the activation energy of the reaction $E_a$[40,42]. $E_{bind}$ determines the
equilibrium constant between products and reactants with
$K = (A_f/A_b) \exp(E_{bind}/k_B T)$, signifying the chemical reversibility
experimentally controlled with regulator[35,43]. It is noted that the small
value of $E_{bind}$ means the strong reaction reversibility.

The effect of chemical reversibility on the crystallization quality of
products was investigated by varying $E_{bind}$. Figure 2e shows the cor-
relation between R and $Q_c$, wherein R is the radius of different spheres,
and $Q_c$ is defined as the ratio of the monomers in six-membered rings
to the total of monomers. The order degree increases along with the
spherical radius, implying that the crystallinity from the inner to the

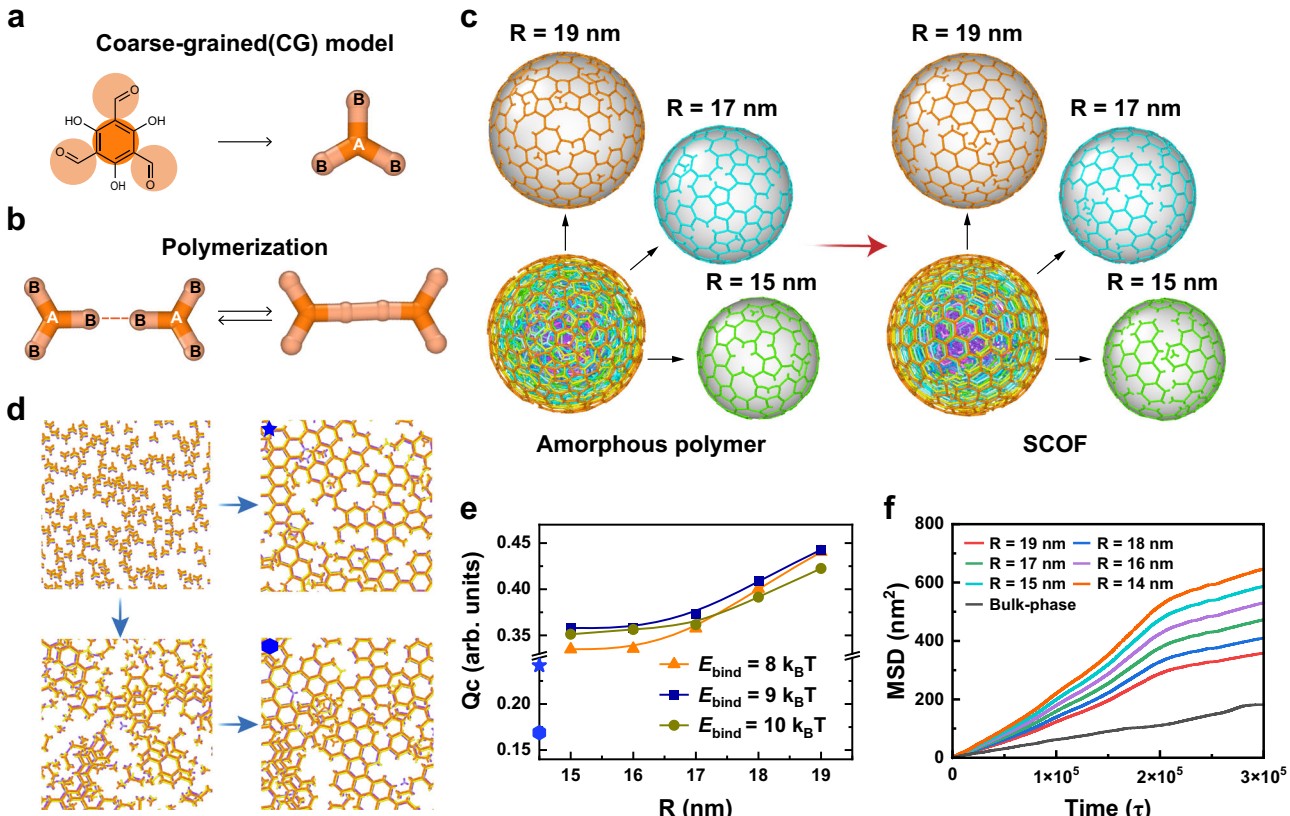

**Fig. 2 | CG-MD simulation for the COF crystallization via the different two pathways. a** Coarse-grained model of monomer Tp. **b** Model reaction forming one B-B bond and two A-B-B angles. **c** Snapshots of spherical COFs with different radiuses formed by a two-step amorphous-to-crystalline process. **d** Snapshots of COFs formed by one-step (marked with a blue star) and two-step (marked with a blue hexagon) methods in the bulk phase, respectively. **e** Crystallization quality (Qc) as a function of the radius R of nanosphere at different $E_{bind}$. The blue symbol on the vertical axis corresponds to the Qc of COF in the bulk phase. **f** Mean-square displacement (MSD) of monomers over time in the crystalline transfer of spherical and bulk COFs. Source data are provided as a Source Data file.

outer sphere is gradually improved in the presence of regulators. This well agrees with the simulation snapshots shown in Fig. 2c, wherein the ordered domains dominate the largest spherical layer ($R = 19$ nm). Thus, we reason that the surface ordering can be accomplished by the regulator-induced zone crystallization. When $E_{bind} = 9k_BT$, the optimum crystallinity was achieved for the different-sized spherical layers, indicating that an appropriate content of regulators used is essential to the structural rearrangement. Without the existence of regulators ($E_{bind} = 15\ k_BT$), limited crystallinity was observed for the different-sized spherical models, elucidating that the amorphous-to-crystalline growth of SCOFs suffers from the sluggish reaction kinetics (Supplementary Fig. 1). In addition, simulations for the bottom-up crystallization of the bulk COF with regulators revealed that the inner layers have higher crystallinity than the outer layers (Supplementary Fig. 2). This unequivocally reflects the difference of the two crystallization pathways in zone crystalline regulation.

To further understand the regulator-enhanced crystallinity for the spherical surface, mean square displacement (MSD) was used to study the diffusion behaviors in crystallization (see the calculation details in the Supplementary Information). Figure 2f exhibits the time-dependent MSD evolution for the bulk phase and spherical phase with different radiuses. The CG model diffusion in the bulk phase is significantly suppressed owing to the lack of a large solid-liquid interface. Accordingly, the molecular rearrangement in the bulk phase is impaired for the regulator-induced zone crystallization (Fig. 2d). In contrast, the molecular diffusion is much easier surrounding the spherical phase and the outer layer of $R = 19$ nm is the most favorable for the free molecular motion. We reason that the layered interaction and steric hindrance lead to the confined diffusion inside the nanosphere, thereby

compromising the inner structural rearrangement. Compared to the one-step crystallization, the two-step method allows for weakened interlayer interactions by solvents, looser space for monomer motion, and larger solid-liquid interfaces for crystal growth. Therefore, the regulator-induced amorphous-to-crystalline transformation is desirable to the targeted zone crystallization, resulting in an ideal COF-based photocatalyst with surface-enhanced electronic properties.

## Synthesis of spherical COFs with enhanced surface crystallinity

Inspired by the MD simulation, we experimentally studied control over the surface crystallinity of COF microspheres by the regulator-mediated amorphous-to-crystalline transformation method (Fig. 3a)[34]. As reported in our early work, the reflux-precipitation polymerization of *p*-phenylenediamine (Pa) and 2,4,6-triformylphloroglucinol (Tp) was carried out in ethanol, rapidly forming the imine-linked amorphous microspheres (TpPa-Polymer) by the Schiff-base reaction. Then, the post-crystallization of TpPa-polymer into TpPa-SCOF was performed under the solvothermal conditions. Such a solid-to-solid structural rearrangement was allowed due to the reversible transamination of imine linkages occurring on the backbones of TpPa-polymer. The resulting TpPa-SCOF almost maintained the original size and morphology, accompanied by the tautomerization from enol-imine to keto-enamine linkage. The surface crystallinity of the microsphere was increased by adopting aniline (An) as a regulator, which was covalently attached to the surface of the primary microsphere (TpPa-Polymer-An) during the precipitation polymerization. After the post-solvothermal treatment, the resulting TpPa-SCOF-An was expected to bear the increasing crystallinity of the peripheral moieties, as predicted by the dynamic simulations.

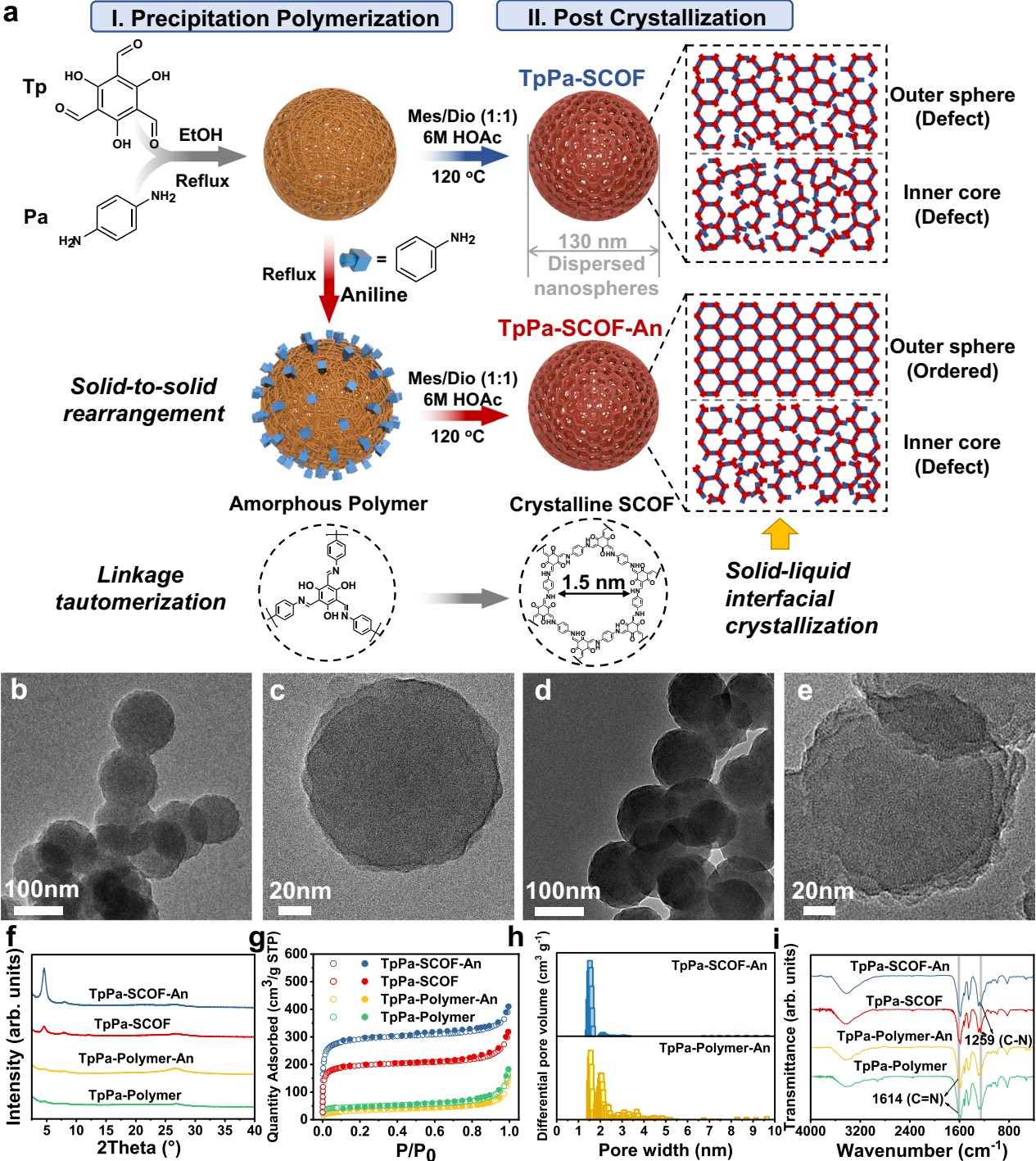

**Fig. 3 | Synthesis and characterization of TpPa-SCOF-An. a** Synthesis of TpPa-SCOF and TpPa-SCOF-An via the amorphous-to-crystalline conversion route, respectively. **b–e** TEM images of (**b**, **c**) TpPa-Polymer-An and (**d**, **e**) TpPa-SCOF-An. **f** PXRD patterns, (**g**) nitrogen adsorption (open circle) and desorption (solid circle) isotherm profiles, **h** pore-size distributions, and (**i**) FT IR spectra of spherical TpPa-polymers and TpPa-SCOFs, respectively. Source data are provided as a Source Data file.

The spherical morphology of products was unequivocally proved by TEM and SEM. As shown in Fig. 3b–e (Supplementary Figs. 3, 4), both TpPa-Polymer-An and TpPa-SCOF-An appear uniformly spherical shapes and narrowly distributed sizes around 130 nm. The reflux-precipitation polymerization determines the spherical morphology and specific particle size, which are still maintained after the subsequent solvothermal treatment. The magnified view of the TpPa-SCOF-An surface revealed the distinct ordered domains, indicative of the high crystallinity, which was in stark contrast to that of TpPa-SCOF without An regulation (Fig. 3e and Supplementary Fig. 5). In addition, as the An moiety was capped on

the surface of COF microspheres, the cross-linking of inter-particles were significantly suppressed, allowing the discrete dispersion.

The An-induced amorphous-to-crystalline transformation was examined by powder X-ray diffraction (PXRD) and $N_2$ sorption measurement. As displayed in Fig. 3f, there are no X-ray diffraction peaks observed for the amorphous TpPa-Polymer-An microsphere, while the crystalline TpPa-SCOF-An confers the typical PXRD pattern as reported for the hexagonal 2D COFs. The (100) peak of TpPa-SCOF-An is relatively higher than that of TpPa-SCOF, signifying that the introduction of An regulator improves the ordered arrangement. Also, the feeding

quantity of the regulator plays a crucial role in determining the eventual crystallinity (Supplementary Fig. 6). This quantity, when appropriately balanced, can facilitate the self-correction of surface defects by being sufficiently exchanged with amino monomers. However, excessive regulators (>16 *equiv.*) lead to a significant decrease in crystallinity and yields as the reaction equilibrium shifts to the framework decomposition (Supplementary Table 1 and Supplementary Fig. 7)[44]. Indeed, TEM images showed that the obtained SCOFs became rougher in surface texture and smaller in size with the An varied from 8-200 *equiv.*, displaying an outer-to-inner decomposition trend (Supplementary Fig. 8). With all in mind, we reason that the surface structural evolution via the An regulation is faster than that for the interior moiety, thereby noticeably promoting crystallinity compared to the inner structure. On the other hand, when COF microspheres are etched using an aqueous solution of NaOH, the ordered layered structures in the crystalline regions may slow the etching process compared to the amorphous regions, resulting in a difference in their appearance. Following alkali post-treatment, TpPa-SCOF-An exhibited internal cavities or slits, while TpPa-SCOF appeared solid on the inside and rough on the surface, resembling a collection of small grains (Supplementary Fig. 9). This contrast likely arises because the increased surface ordering of TpPa-SCOF-An leads to slower etching of the outer components. Although the observed morphologies differ, the remaining components of both samples have identical compositions, with a significant reduction in crystallinity noted after etching (Supplementary Fig. 10).

The porosity of TpPa-SCOF and TpPa-SCOF-An microspheres was assessed by the Brunauer–Emmet–Teller surface areas, reaching 628 and 933 $m^2 g^{-1}$, respectively, both which were far larger than those of the corresponding amorphous analogues (157 and 112 $m^2 g^{-1}$) (Fig. 3g). The pore-size distributions derived from the non-local density functional theory method were populated at 1.5 nm, reflecting a micropore character and ordered pore channels (Fig. 3h).

A variety of spectroscopic techniques were applied to identify the compositions. The FT IR spectra of the amorphous polymers showed the characteristic stretching bands of C=N bond at 1614 $cm^{-1}$, indicative of the formation of imine linkage (Fig. 3i), After transformation, the C=N vibration band was evidently attenuated, and the newly formed C-N bond appeared at 1254 $cm^{-1}$, signifying the imine-to-enamine tautomerization for the resulting β-ketoenamine-linked COF. Again, X-ray photoelectron spectroscopy (XPS) was employed to validate the linkage isomerization (Supplementary Fig. 11). The high-resolution N 1$s$ spectrum of TpPa-SCOF-An was deconvoluted into two individual peaks at 398.4 and 399.4 eV, which were attributable to the imine N and enamine N, respectively. The distinctive C-NH signal manifests the relatively efficient enol-to-keto conversion. Also, it is worth noting that the presence of imine bonds in the amorphous polymer precursor ensures high reversibility and benefits crystallization for TpPa-SCOF-An.

As a control, we prepared TpPa-COF-An using the An-mediated bottom-up route in typical solvothermal conditions. The product appeared the various lengths of nanorods with enhanced crystallinity (Supplementary Figs. 12–15). It is remarkable that mixing the An with monomers in the reaction exerts an impact on the bulk crystallization rather than zone ordering arrangement.

## Photocatalytic performances of TpPa-SCOF-An

The energy band structures of TpPa-SCOF-An, TpPa-SCOF, TpPa-COF-An, and TpPa-COF were investigated prior to the visible photocatalytic test. UV-vis reflectance spectra (UV-vis DRS) of the four COFs were recorded in the solid state (Fig. 4a). The absorption was extended until 900 nm with an onset of 636 nm for all the samples. The optical band gaps were determined to be 2.10, 2.10, 2.06, and 2.07 eV for TpPa-SCOF-An, TpPa-SCOF, TpPa-COF-An, and TpPa-COF, respectively, using the Tauc plots (Fig. 4b). The Mott-Schottky plots suggested the

*n*-type semiconductor character showing the flat band position of − 0.46,− 0.48,− 0.48, and − 0.47 V (*vs*. RHE), respectively (Supplementary Fig. 16). The flat band positions could be roughly regarded as conduction band (CB) levels[45]. Combining the optical band gap and CB value, the valence band (VB) positions were calculated at 1.64, 1.62, 1.58, and 1.60 V (*vs*. RHE), for TpPa-SCOF-An, TpPa-SCOF, TpPa-COF-An, and TpPa-COF, respectively. Taking all into account, the energy band diagrams in Fig. 4c exhibit that all the COFs satisfy the thermodynamic requirements for photocatalytic $H_2$ evolution. In addition, the water vapor uptake measurement verified that all the COFs possessed similar water adsorption capability and wettability (Supplementary Figs. 17, 18).

The photocatalytic hydrogen evolution experiments were performed under visible irradiation (λ > 420 nm) using ascorbic acid as a sacrificial electron donor and Pt as a cocatalyst (Supplementary Fig. 19). Upon exposure to 4-h visible irradiation on TpPa-SCOF-An, hydrogen evolution remained linearly rising (Fig. 4e). The highest hydrogen evolution rate (HER) reached 630 µmol h$^{-1}$ (126 mmol h$^{-1}$ g$^{-1}$, 5 mg COF) when 0.7 wt% Pt was loaded. The performance could be reproduced under identical photocatalytic conditions for TpPa-SCOF-An synthesized from three different batches (Supplementary Fig. 20). Without tedious molecular design and synthesis, TpPa-SCOF-An can be ranked among the top of the reported COF photocatalysts (Supplementary Table 2). The exact loading amount of Pt nanoparticles was as low as 0.14 wt%, as confirmed by the inductively coupled plasma (ICP) spectrometer when feeding the 0.7 wt% Pt precursor. Such an ultralow content of noble metal deposited onto photocatalysts is responsible for a maximum HER for TpPa-SCOF-An (Fig. 4d), manifesting an exceptionally high atom-utilization efficiency in the photochemical reaction. With the added identical Pt precursor (0.7 wt%), the control samples of TpPa-SCOF, TpPa-COF-An, and TpPa-COF deposited 0.29 wt%, 0.55 wt%, and 0.64 wt% of Pt nanoparticles, respectively, while achieving a decrease in the HER of 451, 203, and 165 µmol h$^{-1}$ (Fig. 4e, f). Although the deposited Pt contents for the control samples were adjusted to be 0.1-0.2 wt% similar to that of TpPa-SCOF-An, the corresponding HER values were reduced continuously (Supplementary Fig. 21). Therefore, our study unveils that the cocatalyst Pt is not a leading contribution to the photocatalytic performance of TpPa-SCOF-An.

Undoubtedly, the miniatured particulates with high surface areas is more favorable than the bulky solid in terms of photocatalysis. Indeed, the observed HER of TpPa-SCOF-An was approximately 3.1 times higher than that of TpPa-COF-An. Without the crystallization regulation, the HER of TpPa-SCOF with moderate crystallinity was 71% of that using TpPa-SCOF-An, which has not been reported previously. The surface crystallinity of TpPa-SCOF-An can be finely tuned by varying the solvothermal reaction time, resulting in the adjustable HER performance (Supplementary Fig. 22). This indicates that precise control over surface ordered structure can directly influence the photocatalytic activity.

To validate the applicability of our strategy, we employed two representative amines, 2,5-diaminopyridine (Py) and benzidine (BD), to fabricate spherical COFs with enhanced surface crystallinity (Supplementary Figs. 23, 24). The photocatalytic HERs of TpPy-SCOF-An and TpBD-SCOF-An achieved remarkable values of 507 µmol h$^{-1}$ and 192 µmol h$^{-1}$, respectively. Both were relatively higher than their spherical counterparts synthesized without An regulators. This substantial improvement in photocatalytic performance highlights the applicability of the zone crystallization strategy for enhancing the photocatalytic activity of COFs, making it a potent approach for developing organic photocatalysts.

To elucidate the crucial role of the COF's peripheral moiety on the solid-liquid interfacial photocatalysis, a photo-inert material, $SiO_2$, was wrapped with the An-regulated COF shell for estimating the photocatalytic activity. A template-mediated method was applied to prepare

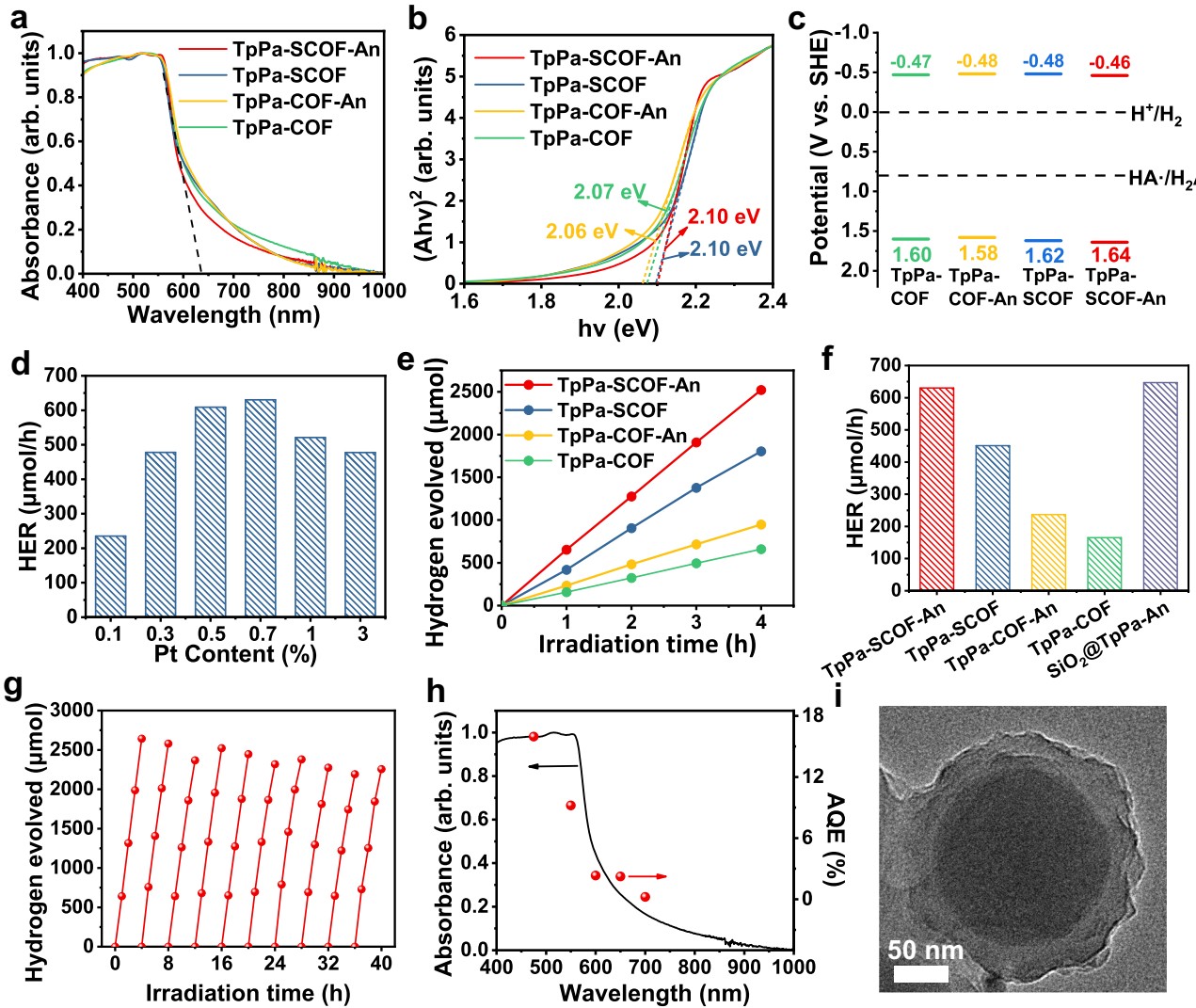

**Fig. 4 | Absorption spectra, band structure, and photocatalytic H₂ evolution.**
**a** UV-vis diffuse reflectance spectra, (**b**) Tauc plots, (**c**) energy-band structures of TpPa-SCOF-An, TpPa-SCOF, TpPa-COF-An, and TpPa-COF, respectively. **d** Photocatalytic performances of TpPa-SCOF-An with different Pt loading amounts. **e** Time course for photocatalytic H₂ production under visible light irradiation. **f** Average H₂ production rates for different photocatalysts. **g** Recyclability of TpPa-SCOF-An for 10 cycles over 40 h under visible light irradiation. **h** Wavelength-dependent AQE of TpPa-SCOF-An superimposed with its absorption curve. **i** TEM image of SiO₂@TpPa-An. Source data are provided as a Source Data file.

a well-defined core/shell microsphere, SiO₂@TpPa-An, consisting of a SiO₂ core (~140 nm) and a TpPa-COF-An shell (~27 nm) by precipitation polymerization and subsequent An-regulated post-crystallization route (Fig. 4i and Supplementary Fig. 25). The outer COF shell accounted for ~37 wt% of the total and remained the characteristic composition and structure of TpPa-COF-An verified by elemental analysis, TGA, PXRD, FT IR, and N₂ sorption (Supplementary Fig. 26 and Supplementary Table 3). The An regulation facilitated the crystallization of the COF shell, showing a relatively stronger (100) lattice signal than that of SiO₂@TpPa synthesized without added An. Under the identical photocatalytic conditions, SiO₂@TpPa-An conferred the similar HER (647 μmol h⁻¹) as TpPa-SCOF-An (630 μmol h⁻¹), demonstrating that the inner COF moiety has analogue to SiO₂ in terms of photocatalysis, both being inactive under visible irradiation (Fig. 4f and Supplementary Fig. 27). The HER per unit weight of COF achieved for SiO₂@TpPa-An is as high as 350 mmol g$_{COF}^{-1}$ h⁻¹. Our findings corroborate that elevating the ordering of outer shells is of significance for optimizing photocatalytic performances.

The photocatalytic recyclability of TpPa-SCOF-An was tested in the presence of PVP as a stabilizer to ensure uniform dispersion over

cycles. There was no significant attenuation observed in H₂ evolution for 10 cycles under 40 h continuous irradiation ($\lambda > 420$ nm), and the linear H₂ evolution curve remained in each cycle (Fig. 4g). The crystallinity, porosity, chemical structure, and light harvesting ability of TpPa-SCOF-An were preserved after photocatalysis as evidenced by PXRD, N₂ sorption, FT-IR, and UV-vis DRS measurements (Supplementary Figs. 28–31). To quantify the conversion of the captured photons, the apparent quantum efficiency (AQE) of TpPa-SCOF-An was evaluated using a few band-pass filters with central wavelengths at 475, 550, 600, 650, and 700 nm, respectively (Fig. 4h). The AQE values were wavelength-dependent and matched well with the absorption curve of TpPa-SCOF-An. The maximum AQE of 15.96% was obtained at 475 nm, outperforming the most reported COF photocatalysts (Supplementary Table 2).

## Photophysical study

To shed light on the origin of the superior photocatalytic activity of TpPa-SCOF-An, we carried out a series of spectroscopic techniques to study the photophysical properties. Figure 5a presents the photoluminescence (PL) emission spectra of different COFs dispersed in

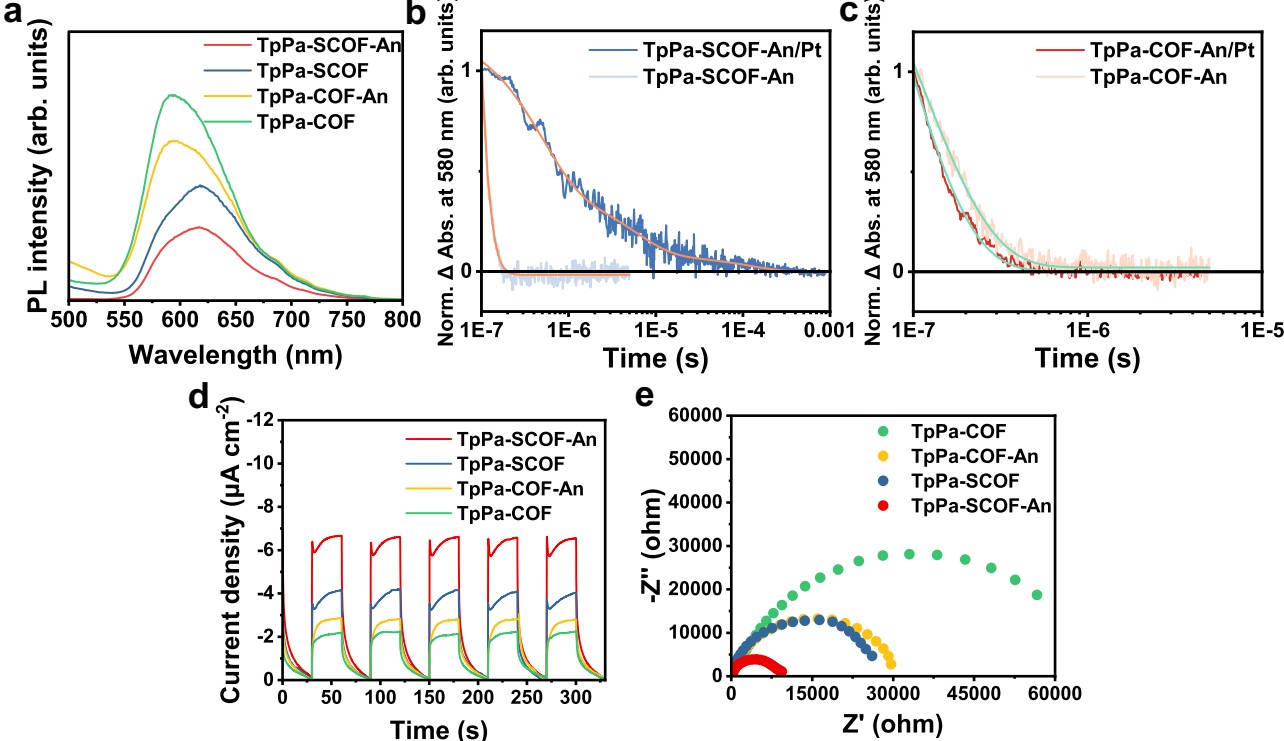

**Fig. 5 | Photophysical measurements. a** Photoluminescent emission spectra ($\lambda_{ex} = 400$ nm) of TpPa-SCOF-An, TpPa-SCOF, TpPa-COF-An, and TpPa-COF, respectively. **b** Normalized transient kinetic decay monitored at 580 nm after pulsed 532 nm excitation for TpPa-SCOF-An and (**c**) TpPa-COF-An, in the presence and absence of the loading of Pt cocatalyst. Bleach signals were normalized to 1 for clear comparison. The overlaid curves in orange and green are kinetic fits to the multi-exponential kinetic model. **d** Transient photocurrent responses and (**e**) Nyquist plots from electrochemical impedance spectroscopy. Source data are provided as a Source Data file.

solution. The PL emission is attenuated with an increase in the COF crystallinity, accompanied with a red shift of emission peak from 594 nm for TpPa-COF and TpPa-COF-An to 618 nm for TpPa-SCOF and TpPa-SCOF-An. It reveals that the expanded crystalline domains weaken the exciton effect and facilitate the charge seperation[46].

The nanosecond transient absorption (TA) spectroscopy was further employed to directly examine the kinetics and dynamics of photogenerated charge carriers in TpPa-SCOF-An and TpPa-COF-An, both with and without Pt cocatalysts loading. The full TA spectra of TpPa-SCOF-An and TpPa-COF-An both showed similar broad excited state absorption (ESA) features, ranging from 350 nm to 800 nm, with decay lifetimes ranging from tens of nanoseconds to around one hundred nanoseconds (Supplementary Fig. 32a, b). The broad ESA band decaying on this time scale was attributed to singlet-singlet annihilation of diffusional excitons[47]. A closer examination of the two full TA spectra revealed a slight downward curvature in the 550–700 nm range for TpPa-SCOF-An relative to TpPa-COF-An, which may arise from the emergence of free charge carriers.

Upon loading with Pt cocatalysts, a prominent ground state bleach (GSB) signal emerged between 550–700 nm in the full TA spectra of TpPa-SCOF-An/Pt, with a dramatically extended lifetime of 98.5 μs, as derived from multiexponential kinetic decay fitting at 580 nm (Supplementary Fig. 32c, Supplementary Table 4, and Fig. 5b). The prominent GSB signal indicates efficient charge separation from exciton singlet-singlet annihilation in TpPa-SCOF-An/Pt as well as free-electron trapping at Pt catalytic sites. The nearly 100 μs long lifetime process is attributed to the charge recombination of free carriers and trapped carriers at Pt catalytic sites. This substantial increase in lifetime, by nearly three orders of magnitude, is beneficial for subsequent H₂ evolution as catalysis typically occurs on much slower time scales. In contrast, the full TA spectra for Pt cocatalysts loaded TpPa-COF-An do not display GSB bands and lack similar enhancements in lifetime,

suggesting inefficient charge separation in TpPa-COF-An (Supplementary Fig. 32b, d, and Fig. 5c). The TA experiments provide direct spectroscopic evidence that the enhanced surface crystallinity in TpPa-SCOF-An helps charge separation and charge recombination, thereby boosting photocatalytic H₂ evolution. The effect of An on regulating surface crystallization is more prominent in the amorphous-to-crystalline transformation than in the bottom-up route, leading to a remarkable increase in surface ordering for long-lived photogenerated active states and robust photocatalytic performance.

Upon switching light on/off, all samples responded to the incident light and TpPa-SCOF-An exhibited the largest photocurrent response (Fig. 5d). Meanwhile, the electrochemical impedance spectroscopy (EIS) demonstrated the smallest semicircle diameter in Nyquist plots for TpPa-SCOF-An, indicative of the minimal charge transfer resistance (Fig. 5e). All the findings can be rationalized by the enhanced surface electronic property of TpPa-SCOF-An.

## Interaction between COF and Pt cocatalyst

As the photocatalytic reaction occurs at the solid-liquid interfaces, exploring the surface electronic structures and chemical reactivity of photocatalysts is essential to disclose the material uniqueness. The peripheral domains of TpPa-SCOF-An feature the enhanced crystallinity containing a large range of periodic microporous frameworks, so the interplay between COF surface and Pt nanoparticles may pronouncedly impact on the reduction reaction efficiency. HR TEM images exhibited the evenly dispersed ultrafine Pt nanoparticles with a diameter of ~2 nm larger than the pore size (1.5 nm), so the deposited Pt mainly resided on the surface of microspheres and the distribution was dominated by periodic atomic frameworks (Fig. 6a, b). As displayed in Fig. 6c, the high-resolution Pt 4$f$ XPS spectrum of TpPa-SCOF-An/Pt can be deconvoluted into the two individual peaks at 72.2 and 75.5 eV, which are assigned to 4$f_{7/2}$ and 4$f_{5/2}$ electrons of Pt(0),

 7

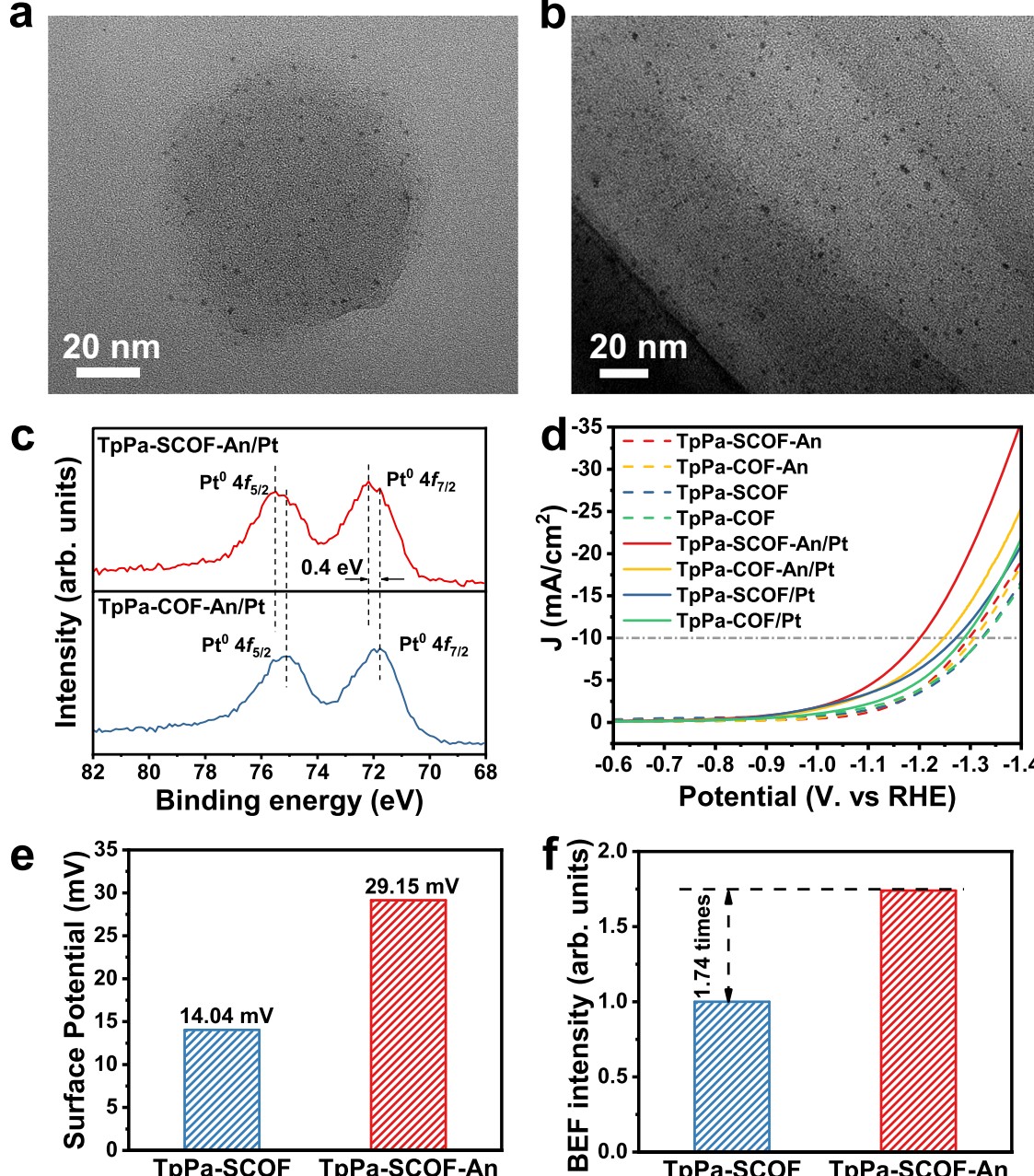

**Fig. 6 | Interaction between SCOF and cocatalyst.** HR TEM images of (**a**) TpPa-SCOF-An/Pt and (**b**) TpPa-COF-An/Pt. **c** High-resolution XPS spectra of Pt 4*f* for TpPa-SCOF-An/Pt and TpPa-COF-An/Pt. **d** LSV plots for photocatalysts before and after Pt loading. **e** Surface potential of TpPa-SCOF-An and TpPa-SCOF. **f** Calculated built-in electric field (BEF) of TpPa-SCOF-An and TpPa-SCOF. Source data are provided as a Source Data file.

respectively. Compared to TpPa-COF-An/Pt, TpPa-SCOF/Pt and TpPa-COF/Pt, the XPS peak position of Pt 4*f* in the TpPa-SCOF-An/Pt was positively shifted by 0.4 eV (Supplementary Fig. 33)[48–50]. The findings manifest that the peripheral crystalline framework with less structural defects is strongly interacted with Pt by the abundant π-electrons, leading to the change of vacant 5*d* band orbital of Pt[50,51]. Accordingly, the Pt nanoparticles could efficiently extract photogenerated electrons from the surface of COF microspheres for proton reduction.

Next, the interfacial electron transfer was studied by comparing the voltammetry behavior of the materials before and after Pt deposition[52,53]. The COFs all displayed similar voltammetry behaviors with the overpotentials of $-1.298$, $-1.308$, $-1.322$, and $-1.325$ V (*vs.* RHE) for TpPa-SCOF-An, TpPa-COF-An, TpPa-SCOF, and TpPa-COF, respectively (Fig. 6d and Supplementary Table 5). The lowest

overpotential of TpPa-SCOF-An can be explained by the higher crystalline surface promoting the electron transfer. After Pt deposition, the overpotential of the materials all positively shifted by 97, 60, 5, and 37 mV (*vs.* RHE) for TpPa-SCOF-An/Pt, TpPa-COF-An/Pt, TpPa-SOF/Pt, and TpPa-COF/Pt, respectively, indicative of the improved reduction ability of the photocatalysts. The dramatically decreased overpotential of TpPa-SCOF-An/Pt suggests the smallest potential barrier and lowest energy dissipation for electron transfer from the peripheral moiety of the COF microsphere to the locally deposited Pt nanoparticles.

Taking all together, we confer that the increased surface crystallinity can alter a built-in electric field (BEF), a crucial driving force to facilitate the surface photogenerated charge separation and transfer kinetics. The intensity of BEF is positively correlated with the surface potential and Zeta potential of the material[54,55]. Using the Kelvin Probe

Force microscopy (KPFM) to observe the discrete particles, the microscopic surface potential of TpPa-SCOF-An was measured to be 29.15 mV, which was almost twice higher than that of TpPa-SCOF (14.04 mV) (Fig. 6e and Supplementary Fig. 34). Meanwhile, we carried out the surface photovoltage test for the macroscopic surface potentials, which were well in line with the KPFM results (Supplementary Fig. 35)[56]. Then, the Zeta potentials of TpPa-SCOF-An and TpPa-SCOF were confirmed to be − 29.1 and − 21.2 mV, respectively (Supplementary Fig. 36). Through the normalized calculation, the BEF intensity of TpPa-SCOF-An was 1.74 times higher than that of TpPa-SCOF (Fig. 6f). Therefore, the stronger BEF overcome energy barriers to powerfully drives the photogenerated electron flow on the surface of TpPa-SCOF-An microspheres in a large range. All the findings substantiate that the elevated order of COF's surface reduces the interfacial energy barrier between peripheral moiety and cocatalyst, accelerates the BEF-driven electron transfer at the solid-liquid interfaces, and in turn, enables excellent photocatalytic performances.

## Discussion

In summary, we report a targeted zone crystallization of spherical COFs through a regulator-induced amorphous-to-crystalline transformation for optimizing the surface electronic property. The aniline regulators are covalently bonded onto the surface of imine-linked amorphous precursors with controlled sizes and spherical morphology. The amorphous-to-crystalline rearrangement of precursors is modulated by the surface-immobilized aniline in the solvothermal conditions for enhanced surface crystallization. Dynamics simulations manifest that such synthesized COFs feature the inner-to-outer increase of crystallinity due to the enhanced regulator motion at a large solid-liquid interface. Therefore, by loading an ultralow content of Pt nanoparticles as a cocatalyst (0.14 wt%), the resulting photocatalyst exhibits the optimum photocatalytic performance. TpPa-SCOF-An gives HER of 126 mmol $g^{-1}$ $h^{-1}$ and an apparent quantum efficiency of 15.96% at 475 nm. By using the identical An-mediated transformation method, the obtained SiO₂@TpPa-An achieves HER of 350 mmol $g_{COF}^{-1}$ $h^{-1}$ comparable to the top of reported organic photocatalysts. The mechanism study elaborates that the prominent surface ordering plays a key role in photo-induced electron extraction, accumulation, and transfer. Without surface defective sites, the strengthened metal-π interaction lowers the overpotentials of electron transfer from the peripheral COF skeletons to the locally deposited Pt nanoparticles, leading to reduced energy dissipation. Also, the surface crystalline domains generate large-range polarization for the built-in electrical field, boosting the photogenerated electron accumulation at the liquid-solid interface. Therefore, our work opens up a promising avenue for modulating zone crystallization for COFs and sheds light on the significance of surface engineering of organic photocatalysts on solar energy conversion.

## Methods

### Synthesis of TpPa-SCOF-An

The synthesis of TpPa-SCOF-An was carried out using the modified two-step method as reported in our previous work[34]. In the first step, a template-free precipitation polymerization was carried out to prepare TpPa-Polymer-An. Typically, 2,4,6-triformylphloroglucinol (Tp) (14.0 mg, 0.067 mmol) and p-phenylenediamine (Pa) (10.8 mg, 0.100 mmol) were dissolved in 5 mL anhydrous ethanol, respectively. The two solutions were rapidly mixed to allow the reaction to proceed under reflux with magnetic stirring for 3 h. Afterwards, aniline (1, 4, 8, 16, 25, 50, 100, 165, 200, and 300 *equiv.* relative to the amount of Tp) was added to the mixture, and the reaction proceeded for another 12 h. After the reaction, the solvent was removed by rotary evaporation to leave the orange powder (TpPa-Polymer-An). In the second step, the obtained product was subjected to the typical solvothermal treatment. A Pyrex tube (10 mL) was charged with the orange solid, a mixed solvent of mesitylene and dioxane (1/1 by *vol.*; 2 mL), and aqueous HOAc

solution as a catalyst (6 M, 0.2 mL). After three freeze-pump-thaw cycles, the tube was sealed off and kept at 120 °C in an oven for 3 days. The products were collected by centrifugation, washed with THF several times, and dried at 40 °C under vacuum to give a red powder (TpPa-SCOF-An) in a yield of 80–91%.

### Photocatalytic H₂ evolution

The photocatalytic hydrogen evolution tests were carried out in a Pyrex top-irradiation reaction vessel connected to a glass-closed Labsolar 6 A gas circulation system (Perfect Light, China). For each reaction, 5 mg photocatalyst was dispersed in an aqueous solution of 0.1 M ascorbic acid and 3.86 mM H₂PtCl₆ aqueous solution was added for photo-deposition of Pt as cocatalyst. Specifically, the added amounts of H₂PtCl₆ aqueous solution (3.86 mM) were 6.8, 20, 34, 47, 68, and 202 μL, corresponding to the feeding Pt contents in Fig. 3d. The mixture was sonicated for 30 min to homogenize the dispersion and then was evacuated several times to remove air completely. The 300 W Xe lamp equipped with a cut-off filter (>420 nm) irradiated on the reaction system through a quartz transparent glass on the top of the vessel. The system was kept at 8 °C by circulating water. The produced gas was analyzed by online GC7900 gas chromatography (Techcomp, China) equipped with a thermal conductivity detector referencing against standard gas with a known concentration of hydrogen. After the photocatalysis test, the samples were recovered by thoroughly rinsing and drying at 40 °C under vacuum.

## Data availability

All data supporting the findings of this study are available within the article, as well as the Supplementary Information file, or available from the corresponding authors on reasonable request. Source data are provided in this paper.

## Code availability

The molecular dynamic simulation script of spherical COF and the analysis data of crystallinity generated in this study have been deposited in the Git Hub under accession code [https://github.com/WreckingKK/NC_SCOF].

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

## Acknowledgements

This work is supported by the National Natural Science Foundation of China (Grants Nos. 52173197 to J.G., 52373202 to J.G., and 22173022 to K.H.).

## Author contributions

J.G. conceived the research and revised the manuscript. Z.L. performed the synthetic experiments, conducted photocatalytic measurements, and wrote the manuscript. X.Y. and Y.Z. performed the MD calculations. Z.Z. and K.H. performed the nanosecond transient absorption measurement. N.D. assisted in the experiments and characterizations. C.W. assisted in the manuscript preparation.

## Competing interests

The authors declare no competing interests.
