## [Peer Review File · Nature Communications]

Controlling crystallization in covalent organic frameworks to facilitate photocatalytic hydrogen production

Corresponding Author: Professor Jia Guo

Version 0:

Reviewer comments:

Reviewer #1

(Remarks to the Author)

In this study, Guo and colleagues proposed a zone crystallization strategy for covalent organic frameworks (COFs) aimed at enhancing the surface ordering of spherical COFs through regulator-induced amorphous-to-crystallinity transformation. They achieved record-high catalytic performance in visible photocatalytic hydrogen evolution. Mechanistic studies shed light on the exceptionally high performance by accumulating photogenerated electrons and lowering the interfacial electron transfer barriers between COF and Pt cocatalysts. This work introduces an intriguing strategy, but a few questions must be addressed before acceptance.

1. The rationale for using aniline in the synthesis is not clearly elucidated. Additionally, the covalent connection of aniline onto the COF requires further characterization.
2. The mechanism of amorphous-to-crystalline transformation in keto-enamine-linked COFs remains unclear. Tautomerization converts the reversible imine into irreversible ketoenamine, "locking" the dynamic bonds. If so, how does the amorphous-to-crystalline transformation occur in TpPa-SCOF? Will it bypass the bond reforming step?
3. The author did not denote TpPa-COF-An, making reading challenging.
4. Table S2 missed notable COF-based HER photocatalysts:
 - a. Nat. Commun., 2022, 13, 2357
 - b. J. Am. Chem. Soc., 2023, 145, 15, 8364–8374
 - c. Nature, 2022, 604, 72–79
 - d. Adv. Funct. Mater., 2022, 32, 2207394
5. "The HER of TpPa-SCOF with inner high crystallinity was 71% of that using TpPa-SCOF-An with outer high crystallinity." Why does TpPa-SCOF possess inner high crystallinity? Scheme 1 indicates both COFs have low inner crystallinity.
6. Why does SiO₂ promote the HER performance?
7. Another control, SiO₂@TpPa, is needed to showcase the highly ordering outer shell's essential nature.

Reviewer #2

(Remarks to the Author)

The manuscript titled "Zone Crystallization modulation of Covalent Organic Frameworks Enhancing Surface Electronic Property for Photocatalytic Hydrogen Evolution" presents an innovative zone crystallization strategy for enhancing the surface ordering of spherical covalent organic frameworks (COFs) through regulator-induced amorphous-to-crystallinity transformation. The study firstly utilizes dynamic simulations to elucidate the pivotal role of surface-bound regulators in enhancing surface dynamic reversibility, which fosters a crystallinity gradient from the inner to the outer layers, contrasting with conventional bottom-up COF growth methods. The authors further demonstrated that enhanced surface crystallinity not only augments backbone polarization, thereby constructing surface electrical fields to accumulate photogenerated electrons, but also significantly reduces the electron transfer barriers between the COF skeletons and Pt cocatalysts. As a result, the visible-light-driven photocatalytic hydrogen evolution rate achieved is notably high, recording 126 mmol g⁻¹ h⁻¹ for the simplest β-ketoenamine-linked COF (TpPa, phenylene as a linker) and a record-high rate of 350 mmol gCOF⁻¹ h⁻¹ for a core/shell-structured SiO₂@COF(TpPa), using an ultralow 0.7wt% Pt as a cocatalyst. Overall, the manuscript is well-written and makes a significant advancement in the preparation and application of COFs-based photocatalysts. I would like to

recommend the publication in Nature Communications after addressing the following minor concerns:

- (1) The authors mentioned that the surface crystallinity is rather weaker than the inner for the conventional polycrystalline COFs solids, it is suggested to include some references for this statement.
- (2) Besides the dynamic simulations, is there any experimental data to support the difference of inner- and outer-crystallinity? Could the authors employ high-resolution transmission electron microscopy (HR-TEM) to visually confirm this gradation in crystallinity?
- (3) In Fig.2, please ensure that all TEM images include a uniform scale bar to facilitate direct comparison.
- (4) It is interesting to notice that the exact loading amount of Pt by ICP were different for TpPa-SCOF-An-Pt, TpPa-SCOF-Pt, TpPa-COF-An-Pt, and TpPa-COF-Pt, respectively. Will that affect the results of this manuscript? It is also suggested to describe how to control the Pt loading in Fig. 3d.
- (5) It is suggested to include SEM characterization for all COFs, particularly to observe the morphology and distribution of nanoparticles.
- (6) Considering that the HER for the SiO₂@TpPa-An was a very high number, it is necessary to confirm the TaPa content carefully beside the TGA measurement. It is also suggested to include the TGA profile of TpPa-An for a comparative assessment with SiO₂@TpPa-An and SiO₂.
- (7) For the supplementary table 2, more references should be covered, especially more recent papers reporting high HER.
- (8) For the second step of preparation, 3 days was used. By varying the reaction time, will the crystallinity be finely tuned? The time-dependent experiments might be studied for the effect of crystallinity on the HER performance.
- (9) The page number was missing for ref. 2.

Reviewer #3

(Remarks to the Author)

This manuscript reported a zone crystallization strategy for COFs to enhance the surface ordering of spherical COFs through the regulator-induced amorphous-to-crystallinity transformation. The visible photocatalytic hydrogen evolution rate can reach an exceptionally high value of 126 mmol g⁻¹ h⁻¹. However, the connection between the structure of the materials synthesized in the manuscript and the synthesis strategy remains unclear. It is not evident from the paper whether materials with similar structures can be consistently synthesized under identical conditions, as there is a lack of supporting replication data. Moreover, the manuscript contains many problems as I mentioned below. The revised version can be published in Nature Communications.

1. The author should clearly specify what the regulator is in the abstract.
2. How is the shape of COF controlled to be spherical? Has only this type of COF (TpPa-SCOF) been studied? Has there been any research on the effects of the strategies proposed in the article on the surface crystallinity and catalytic activity of other types of COFs? Is this strategy universally applicable?
3. It is recommended to use HRTEM (High-Resolution Transmission Electron Microscopy) to provide further evidence of the improvement in surface crystallinity.
4. As we can see from Supplementary Fig.8, the crystallinity of TpPa-COF is also remarkably enhanced with assistance of An regulator. Why the author claim that "the localized ordered domains within the outer COF solids is not purposefully increased"? The author should provide more evidence.
5. How can the authors determine if the enhancement in catalytic activity of COFs after the addition of aniline is due to increased surface crystallinity? Is it related to an increase in surface area? Which has a greater impact, crystallinity or surface area?
6. What are the Pt contents in the catalysts shown in Fig. 5? This is crucial for comparing their reaction activities.
7. The effect of growing pathways on zone crystallization of COF in Scheme 1 are quite confusing. Could you use a clearer layout to better convey the differences between these two methods?
8. More relevant references about the COFs crystallinity and photocatalytic performance should be cited, e.g. DOI: 10.1038/s41586-022-04443-4; 10.1038/s41467-023-42757-7,;
9. The differences in the high-resolution XPS spectra between Pt 4f for TpPa-SCOF -An/Pt and TpPa-COF-An/Pt are not obvious. The statement "The findings manifest that the outer crystalline framework with less structural defects is strongly interacted with Pt by the abundant π -electrons leading to the change of vacant 5d band orbital of Pt." lacks sufficient evidence.
10. Did the author experiment with the maximum concentration limit of aniline, beyond which COFs cannot be formed? Please report on this.

Reviewer #4

(Remarks to the Author)

In the present manuscript, Lin and co-workers have reported the effect of nanoscale crystallinity over the photocatalytic hydrogen evolution, which they have denoted as the "zone crystallization". The crystallinity of the COF nanoparticles was tuned by using a modulator, leading to the superhigh rate of photocatalytic hydrogen evolution. After reading the manuscript carefully, I find the manuscript lacks novelty that required for publishing in Nature Communications for the following reasons:

- a) Transforming kinetically controlled amorphous polymer into thermodynamically stable crystalline COFs is well documented COF chemistry. Also, the usage of modulator to tune or improve the crystallinity has been extensively reported by various research groups.
- b) Although the Scheme 1 is visually appealing and convincing, the manuscript fails to provide experimental evidence that support the hypothesis. Specifically, it is unclear how the authors distinguish experimentally between localized and delocalized surface carriers.

c) While it is fundamentally established that a built-in electric field helps to separate the charge carriers, however, the manuscript does not clarify how crystalline COF has high built-in electric field compared to the amorphous counterpart. I am sceptical about the KPFM results, due to the lack of detailed methodology, including sample preparation and experimental conditions, moreover the measurement was conducted for a single nanoparticle. To validate the result, more than 100 COF particles should be screened.

d) The reported rate of photocatalytic hydrogen evolution is exceptionally high, which need to be verified. While, I do not doubt the author's results, however previously, it was noticed several instances that the photocatalytic activity varied significantly under different experimental conditions.

Version 1:

Reviewer comments:

Reviewer #1

(Remarks to the Author)

I commend the authors for their significant effort in addressing the feedback I provided. The manuscript has been substantially improved as a result. Based on these improvements, I support the acceptance of this work for publication

Reviewer #2

(Remarks to the Author)

I think that the authors have well addressed the comments made by the reviewers in the revised version of the manuscript. Therefore, I have no further questions. It should be accepted.

Reviewer #3

(Remarks to the Author)

I have carefully reviewed the revised version of the manuscript and Point-by-point Response to the Comments. Basically met the revision requirements and agreed to be published.

Reviewer #4

(Remarks to the Author)

I have gone through the revised manuscript and response letter. However, still I am not convinced by the author's justifications, because of the following reasons.

a) the concept of zone crystallization is not validated as shown in scheme 1, and it does not match with TEM image in SI fig 5. It's still more of an amorphous to crystalline transformation.

b) Regarding delocalized surface carriers, I do not agree with the authors because the long lifetime of the photogenerated electron can be attributed to the trapped electrons. Also, if I consider the kinetic fits, long lifetime component has very little population.

Overall, the claims are too ambitious compared to the experimental evidence.

Version 2:

Reviewer comments:

Reviewer #4

(Remarks to the Author)

Although I do not completely agree with the claims of the present manuscript, however the authors have done a large number of supporting experiments to indirectly prove their claims. Authors are requested to address the following comments

a) Etching with NaOH cannot selectively etch the amorphous region because principally both crystalline and amorphous domain can have imine and enamine linkage. Also, authors did not perform any analytical experiments after the etching with NaOH, such as PXRD, IR and solid state CP MAS NMR, which will provide detail information about the structure.

b) Delocalized charge carries should reach the Pt centers quickly and leading to fast kinetics however here we observed different trends in TAS analysis.

Version 3:

Reviewer comments:

Reviewer #4

(Remarks to the Author)

The authors have addressed all the comments and now the manuscript can be accepted.

Reviewer #1 (Remarks to the Author):

In this study, Guo and colleagues proposed a zone crystallization strategy for covalent organic frameworks (COFs) aimed at enhancing the surface ordering of spherical COFs through regulator-induced amorphous-to-crystallinity transformation. They achieved record-high catalytic performance in visible photocatalytic hydrogen evolution. Mechanistic studies shed light on the exceptionally high performance by accumulating photogenerated electrons and lowering the interfacial electron transfer barriers between COF and Pt cocatalysts. This work introduces an intriguing strategy, but a few questions must be addressed before acceptance.

1. The rationale for using aniline in the synthesis is not clearly elucidated. Additionally, the covalent connection of aniline onto the COF requires further characterization.

Response: Thanks for your issue. As early reported (*Ref. 35 T. Ma, et al. Science, 2018, 361, 48*), the purpose of using aniline in synthesis is to improve the crystallinity of COF through the reversible transamination reaction between aniline and amine monomers. Based on this modulation strategy, we covalently attached aniline on the surface of amorphous frameworks, so the surface amorphous-to-crystalline transformation can be intensified by the aniline-mediated transamination, thereby achieving the higher crystallinity outside than inside.

The covalent connection of aniline onto the COF is hard identified as the formed linkage was the same with the main backbone. To demonstrate the presence of aniline on the COF, we used sodium sulfanilate as a regulator to synthesize the TpPa-SCOF-SO₃Na via the two-step crystallization under the identical conditions. As shown in **Fig. R1**, the characteristic stretching bands of S=O and S-O bonds at 1122 and 1035 cm⁻¹ derived from sodium sulfanilate can be observed on the FTIR spectrum of TpPa-SCOF-SO₃Na. This reflects that the aniline regulator can remain on the COF matrix during the amorphous-to-crystalline conversion.

Fig. R1 FT IR spectra of sodium sulfanilate, TpPa-SCOF-SO₃Na, and TpPa-SCOF.

2. The mechanism of amorphous-to-crystalline transformation in keto-enamine-linked COFs remains unclear. Tautomerization converts the reversible imine into irreversible ketoenamine, "locking" the dynamic bonds. If so, how does the amorphous-to-crystalline transformation occur in TpPa-SCOF? Will it bypass the bond reforming step?

Response: Thanks for your issue. The amorphous-to-crystalline transformation relies on the reversible Schiff-base reaction. Specifically, the first-step synthesis is the precipitation polymerization of Pa and Tp under reflux, forming the imine-linked amorphous frameworks, which has been proven by FT IR spectra in **Fig. 2i**. Then, the structural transformation from amorphous frameworks to crystalline COF was allowed under solvothermal conditions, accompanied by the conversion of enol-imine to keto-enamine linkages. Therefore, the imine-linked amorphous precursor ensures the occurrence of reversible transformation into thermodynamically stable keto-enamine-linked COF. The mechanism has been also well demonstrated by our group in the early work (*Ref. 33, J. Tan, et al. Angew. Chem. Int. Ed. 2016, 55, 13979*).

The related discussion has been revised in the main text, *Page 9, Line 7-16*, "...the reflux-precipitation polymerization of *p*-phenylenediamine (Pa) and 2,4,6-triformylphloroglucinol (Tp) was carried out in ethanol, rapidly forming the imine-linked amorphous microspheres (TpPa-Polymer) by the Schiff-base reaction. Then, the post-crystallization of TpPa-polymer into TpPa-SCOF was performed under the solvothermal conditions. Such a solid-to-solid structural rearrangement was allowed due to the reversible transamination of imine linkages occurring on the backbones of TpPa-polymer. The resulting TpPa-SCOF almost maintained the original size and morphology, accompanied by the tautomerization from enol-imine to keto-enamine linkage."

3. The author did not denote TpPa-COF-An, making reading challenging.

Response: Thanks for your reminder. TpPa-COF-An is a control sample synthesized through a direct solvothermal method using aniline as a regulator, but without control over the COF morphology and sizes.

The related illustration has been added in the main text, *Page 12 Line 6-8*, "As a control, we prepared TpPa-COF-An using the An-mediated bottom-up route in typical solvothermal conditions. The product appeared the various lengths of nanorods with enhanced crystallinity (**Supplementary Figs. 10-13**)."

4. Table S2 missed notable COF-based HER photocatalysts:

- a. Nat. Commun., 2022, 13, 2357
- b. J. Am. Chem. Soc., 2023, 145, 15, 8364–8374
- c. Nature, 2022, 604, 72–79
- d. Adv. Funct. Mater., 2022, 32, 2207394

Response: Thanks for your reminder. According to the suggestion, we have cited the references above in **Supplementary Table 2**.

The added references are as follows.

“[S23] Li, Z. et al. Three-Component Donor- π -Acceptor Covalent-Organic Frameworks for Boosting Photocatalytic Hydrogen Evolution. *J. Am. Chem. Soc.* 145, 8364–8374, doi:10.1021/jacs.2c11893 (2023).

[S28] Zhang, W. et al. Reconstructed covalent organic frameworks. *Nature* 604, 72-79, doi:10.1038/s41586-022-04443-4 (2022).

[S30] Li, C., Liu, J., Li, H. et al. Covalent organic frameworks with high quantum efficiency in sacrificial photocatalytic hydrogen evolution. *Nat. Commun.* 13, 2357, doi:10.1038/s41467-022-30035-x (2022).

[S31] Li, W., et al. Tuning Molecular Chromophores of Isoreticular Covalent Organic Frameworks for Visible Light-Induced Hydrogen Generation. *Adv. Funct. Mater.*, 32, 2207394, doi:10.1002/adfm.202207394 (2022)”

The above references are added in the *Supplementary Table 2*.

Catalyst	Co-catalyst	Sacrificial agent	HER ($\mu\text{mol g}^{-1}\text{h}^{-1}$)	AQE (%)	Ref.
TCDA-COF (COF-JLU35)	Pt	Ascorbic acid	70,800	2.57 (450 nm)	[S23]
RC-COF-1	Pt	Ascorbic acid	27,980	6.39 (420 nm)	[S28]
CYANO-CON	Pt	Ascorbic acid	134,200	82.6 (450 nm)	[S30]
USTB-10	Pt	Ascorbic acid	21,800	0.68 (420 nm)	[S31]

5. "The HER of TpPa-SCOF with inner high crystallinity was 71% of that using TpPa-SCOF-An with outer high crystallinity." Why does TpPa-SCOF possess inner high crystallinity? Scheme 1 indicates both COFs have low inner crystallinity.

Response: Thanks for your reminder. Here is not an appropriate description. Without aniline regulation, the crystallinity of TpPa-SCOF is moderate due to the slow reaction kinetics for the amorphous-to-crystalline structural rearrangement.

We have revised the sentence in the main text, *Page 14, Line 10-13*, “More significantly, without the crystallization regulation, the HER of TpPa-SCOF with moderate crystallinity was 71% of that using TpPa-SCOF-An, which has not been reported previously.”

6. Why does SiO₂ promote the HER performance?

Response: Thanks for your issue. Incorporating a photo-inert SiO₂ core aims to demonstrate that the inner COF moiety has little contribution to the photocatalytic performance. As shown in Fig.3f, the photocatalytic HER of SiO₂@TpPa-An (647 $\mu\text{mol h}^{-1}$, 5 mg catalyst) is similar to that of TpPa-SCOF-An (630 $\mu\text{mol h}^{-1}$, 5 mg

catalyst), suggesting that the photocatalytic reaction mainly occurred on the outer TpPa-SCOF-An microsphere. The role of the inner COF moiety is akin to the SiO₂ core, both being inactive under light irradiation. These findings have significant implications for our understanding of photocatalysis. As the used photocatalysts kept the equal concentration, the calculated HER of SiO₂@TpPa-An per unit weight of COF is much larger than that of TpPa-SCOF-An, but they have the similar photocatalytic activity.

The change has been made in the main text, *Page 15, Line 16-21*, “Under the identical photocatalytic conditions, SiO₂@TpPa-An conferred the similar HER (647 μmol h⁻¹) as TpPa-SCOF-An (630 μmol h⁻¹), demonstrating that the inner COF moiety has analogue to SiO₂ in terms of photocatalysis, both being inactive under visible irradiation (**Fig. 3f** and **Supplementary Fig. 25**). We underline that the record-high HER per unit weight of COF is achieved for SiO₂@TpPa-An up to 350 mmol g_{COF}⁻¹ h⁻¹.”

7. Another control, SiO₂@TpPa, is needed to showcase the highly ordering outer shell's essential nature.

Response: Thanks for your suggestion. To illustrate the ordering of the COF shell in SiO₂@TpPa-An, we prepared a control sample SiO₂@TpPa without aniline regulation for comparison of crystallinity. SiO₂@TpPa-An exhibited a more intense (100) X-ray diffraction peak than SiO₂@TpPa (**Supplementary Fig. 24a**), manifesting the higher ordering of the COF shell for SiO₂@TpPa-An formed by the aniline-induced structural rearrangement. However, the ultrathin organic shell of less than 30 nm on the SiO₂ core cannot tolerate the high-voltage electron beam, so the ordered pore alignment of COF is destroyed against observation.

The change has been made in the main text, *Page 15, Line 14-16*, “The An regulation facilitated the crystallization of the COF shell, showing the relatively stronger (100) lattice signal than that of SiO₂@TpPa synthesized without added regulators.”

Reviewer #2 (Remarks to the Author):

The manuscript titled "Zone Crystallization modulation of Covalent Organic Frameworks Enhancing Surface Electronic Property for Photocatalytic Hydrogen Evolution" presents an innovative zone crystallization strategy for enhancing the surface ordering of spherical covalent organic frameworks (COFs) through regulator-induced amorphous-to-crystallinity transformation. The study firstly utilizes dynamic simulations to elucidate the pivotal role of surface-bound regulators in enhancing surface dynamic reversibility, which fosters a crystallinity gradient from the inner to the outer layers, contrasting with conventional bottom-up COF growth methods. The authors further demonstrated that enhanced surface crystallinity not only augments backbone polarization, thereby constructing surface electrical fields to accumulate photogenerated electrons, but also significantly reduces the electron transfer barriers between the COF skeletons and Pt cocatalysts. As a result, the visible-light-driven photocatalytic hydrogen evolution rate achieved is notably high, recording 126 mmol g⁻¹ h⁻¹ for the simplest β-ketoenamine-linked COF (TpPa, phenylene as a linker) and a record-high rate of 350 mmol gCOF-1 h-1 for a core/shell-structured SiO2@COF(TpPa), using an ultralow 0.7wt% Pt as a cocatalyst. Overall, the manuscript is well-written and makes a significant advancement in the preparation and application of COFs-based photocatalysts. I would like to recommend the publication in Nature Communications after addressing the following minor concerns:

(1) The authors mentioned that the surface crystallinity is rather weaker than the inner for the conventional polycrystalline COFs solids, it is suggested to include some references for this statement.

Response: Thanks for your suggestion. The reference regarding the weak surface crystallinity have been cited for the statement, i.e., *Ref 23-25*.

The added references are as follows.

“23 Daliran, S. et al. Defects and Disorder in Covalent Organic Frameworks for Advanced Applications. *Adv. Funct. Mater.*, **34**, 2312912. doi: 10.1002/adfm.202312912 (2024).

24 Calik, M. et al. From Highly Crystalline to Outer Surface-Functionalized Covalent Organic Frameworks—A Modulation Approach. *J. Am. Chem. Soc.*, **138**, 1234–1239, doi:10.1021/jacs.5b10708 (2016).

25 Guan, Q. et al. BODIPY-Decorated Nanoscale Covalent Organic Frameworks for Photodynamic Therapy. *iScience*, **14**, 180–198, doi: 10.1016/j.isci.2019.03.028 (2016).”

The discussion with references is added in the main text, *Page 4, Line 10-12*, “For the polycrystalline COF solids that grow up from starting materials, the surface crystallinity is relatively weaker than the inner moiety, with numerous remaining defects and unreactive terminals due to incomplete conversion.²³⁻²⁵”

(2) Besides the dynamic simulations, is there any experimental data to support the difference of inner- and outer- crystallinity? Could the authors employ high-resolution transmission electron microscopy (HR-TEM) to visually confirm this gradation in crystallinity?

Response: Thanks for your valuable suggestion. We adopted HR TEM to observe the TpPa-SCOF-An and TpPa-SCOF. As TEM cannot image through thick samples, it was hard to identify the inner atomic arrangement, but a comparison of the outer crystallinity for TpPa-SCOF-An and TpPa-SCOF is likely. As shown in the HR TEM image (*Supplementary Fig. 5*), the ordered pore alignments are explicitly shown on the surface of TpPa-SCOF-An. In contrast, much fewer crystalline domains were observed on TpPa-SCOF than on TpPa-SCOF-An. This indicates that the surface crystallinity is primarily improved by the regulator induction.

Although it is challenging to observe inner ordering, we attribute the inner-to-outer increase in crystallinity to the intensified surface reaction kinetics. As the surface crystallization proceeds, the surface morphology and particle size should be concurrently changed by the surface-immobilized aniline. Thus, we studied the effect of aniline on the surface morphology, particle size, product yield, and COF crystallinity to clarify whether the aniline significantly alters the surface reaction kinetics (*Supplementary Figs. 6-8 and Supplementary Table 1*).

As the added amount of aniline was varied from 16 to 300 equivalents, we observed a continuous decrease in the product yield and crystallinity, and simultaneously, the resulting COF microspheres became rougher in surface texture and smaller in size. When 300 equivalents of aniline were used, the product decomposed without any solids collected. The findings elucidate that the excessive aniline anchored on the surface essentially impairs the reaction reversibility with amino monomers, leading to surface deformation, size reduction, and even final degradation from outside to inside. It further reveals that the surface-immobilized aniline has a more pronounced impact on the outer reaction kinetics, thus readily achieving a higher crystallinity near the surface than insides.

The related discussion has been added in the main text, *Page 10, Line 18-22 and Page 11, Line 1-8*, “Also, we found that the feeding quantity of the regulator plays a crucial role in determining the eventual crystallinity (**Supplementary Fig. 6**). This quantity, when appropriately balanced, can facilitate the self-correction of surface defects by being sufficiently exchanged with amino monomers. However, excessive regulators (> 16 *equiv.*) lead to a significant decrease in crystallinity and yields as the reaction equilibrium shifts to the framework decomposition (**Supplementary Table 1 and Supplementary Fig. 7**).⁴⁴ TEM observation showed that the SCOF particulates became rougher in surface texture and smaller in size with the An varied from 8-200 *equiv.*, displaying an outer-to-inner decomposition tend (**Supplementary Fig. 8**). These findings indicate that the exchange between An and amino monomers on the COF

surface achieves equilibrium between the growth and decomposition of surface crystalline structure, thereby promising for the higher outer crystallinity compared to the inner core.” And the related synthesis method has been revised in the main text, **Page 21, Line 19-20**, “Afterwards, aniline (1-300 *equiv.* relative to the amount of Tp) was added to the mixture and the reaction proceeded for another 12 h.”

The related figures have been added in the **Supplementary Figs. 5-8** as shown below.

Supplementary Fig. 5 HR TEM images of the surface crystalline structures for (a) TpPa-SCOF-An and (b) TpPa-SCOF. The comparison shows the improved ordered pore alignment on the surface of TpPa-SCOF-An.

Supplementary Fig. 6 PXRD patterns of TpPa-SCOF-An prepared by varying the added amounts of aniline from 1 to 200 equivalent.

Supplementary Fig. 7 Photograph of the post-reaction solution using excessive aniline (300 equiv.) for synthesizing TpPa-SCOF-An.

Supplementary Fig. 8 TEM images of TpPa-SCOF-An prepared with (a) 8 *equiv.*, (b) 25 *equiv.*, (c) 50 *equiv.*, (d) 100 *equiv.*, (e) 165 *equiv.*, and (f) 200 *equiv.* of aniline.

The related table has been revised as below.

Supplementary Table 1. Yields of TpPa-SCOF-An synthesized with different added amounts of aniline under otherwise identical conditions.

Amounts of aniline	Yield
1 equiv.	91.5%
4 equiv.	91.2%
8 equiv.	90.0%
16 equiv.	88.6%
25 equiv.	80.4%
50 equiv.	74.4%
100 equiv.	59.6%
165 equiv.	46.2%
200 equiv.	21.5%
300 equiv.	0

(3) In Fig.2, please ensure that all TEM images include a uniform scale bar to facilitate direct comparison.

Response: According to your suggestion, we have unified the scale bar of the TEM images in **Fig. 2**.

The related figures have been revised in the main text, *Fig. 2*, as shown below.

Fig. 2 TEM images of **b, c** TpPa-Polymer-An and **d, e** TpPa-SCOF-An.

(4) It is interesting to notice that the exact loading amount of Pt by ICP were different for TpPa-SCOF-An-Pt, TpPa-SCOF-Pt, TpPa-COF-An-Pt, and TpPa-COF-Pt, respectively. Will that affect the results of this manuscript? It is also suggested to describe how to control the Pt loading in Fig. 3d.

Response: Thanks for your issue. Through the study on the photocatalytic performance, the feeding Pt precursor was optimized to be 0.7wt% and the exact deposition contents of Pt nanoparticles were identified to be 0.14wt%, 0.29wt%, 0.55wt%, and 0.64wt% for TpPa-COF-An, TpPa-SCOF, TpPa-COF-An, and TpPa-COF, respectively, using ICP spectrometer. TpPa-COF-An achieved a maximum HER value, followed by TpPa-SCOF, TpPa-COF-An, and TpPa-COF in sequence. Thus, the lowest Pt content is responsible for the highest HER, implying that an increase in cocatalyst Pt is adverse to the photocatalytic activity. We also varied the added quantity of Pt precursor to ensure that the deposited contents were 0.1-0.2wt% for all the samples. The HER change trend remained consistent. Therefore, we can confidently state that the difference of photocatalytic activity for those samples are not originate from the co-catalyst influence.

The related discussion has been added in the main text, *Page 13, Line 17-19*, “The exact loading amount of Pt nanoparticles was as low as 0.14wt%, as confirmed by the inductively coupled plasma (ICP) spectrometer when feeding the 0.7wt% Pt precursor.” *Page 13, Line 22* and *Page 14, Line 1-5*, “With the added identical Pt precursor (0.7wt%), the control samples of TpPa-SCOF, TpPa-COF-An, and TpPa-COF deposited 0.29wt%, 0.55wt%, and 0.64wt% of Pt nanoparticles, respectively, while achieving a decrease in the HER of 451, 203, and 165 $\mu\text{mol h}^{-1}$ (**Figs. 3e, 3f**). Although the deposited Pt contents for the control samples were adjusted to be 0.1-0.2 wt% similar to that of TpPa-SCOF-An, the corresponding HER values were reduced continuously (**Supplementary Fig. 19**).”

The description of Pt loading method was added in the main text, *Page 22, Line 13-15*, “Specifically, the added amounts of H_2PtCl_6 aqueous solution (3.86 mM) were 6.8, 20, 34, 47, 68, and 202 μL , corresponding to the feeding Pt contents in Fig. 3d.”

Supplementary Fig. 19 H₂ production rates for the different photocatalysts loaded with the similar Pt amount measured by ICP, which were 0.14, 0.19, 0.16, and 0.12 wt%, respectively, for TpPa-SCOF-An-Pt, TpPa-SCOF-Pt, TpPa-COF-An-Pt, and TpPa-COF-Pt. The results suggested that the observed difference in photocatalytic activity is not due to variation in the exact Pt loading amounts.

(5) It is suggested to include SEM characterization for all COFs, particularly to observe the morphology and distribution of nanoparticles.

Response: Thanks for your suggestion. We have conducted the SEM characterization for TpPa-SCOF-An, TpPa-SCOF, TpPa-COF-An, and TpPa-COF. The SEM images show that both TpPa-SCOF-An and TpPa-SCOF share a spherical shape, while TpPa-COF-An and TpPa-COF are rod-like, a consistent observation with our TEM images. The particle-size distribution was statistically determined by measuring the diameters of 100 particles, of which TpPa-SCOF-An and TpPa-SCOF both predominated around 130 nm.

The change has been made in the main text, *Page 10 Line 1-4*, “The spherical morphology of products was unequivocally proved by TEM and SEM. As shown in **Figs. 2b-e (Supplementary Figs. 3, 4)**, both TpPa-Polymer-An and TpPa-SCOF-An appear uniformly spherical shapes and narrowly distributed sizes around 130 nm.” And *Page 12, Line 6-8*, “As a control, we prepared TpPa-COF-An using the An-mediated bottom-up route in typical solvothermal conditions. The product appeared the various lengths of nanorods with enhanced crystallinity (**Supplementary Figs. 10-13**).”

The related figures have been added in the *Supplementary Figs. 3 and 10*, as shown below.

Supplementary Fig. 4 SEM images and the derived particle size distributions for (a) TpPa-SCOF-An and (c) TpPa-SCOF.

Supplementary Fig. 12 SEM images of (a) TpPa-COF-An and (b) TpPa-COF.

(6) Considering that the HER for the SiO₂@TpPa-An was a very high number, it is necessary to confirm the TaPa content carefully beside the TGA measurement. It is also suggested to include the TGA profile of TpPa-An for a comparative assessment with

SiO₂@TpPa-An and SiO₂.

Response: Thanks for your valuable suggestion. The loaded content of COF in SiO₂@TpPa-An was precisely evaluated by measuring Si contents using ICP-AES. Thus, SiO₂ and SiO₂@TpPa-An afforded 12.73wt% and 8.14wt% of Si contents, respectively, and the corresponding COF(TpPa-An) shell was calculated to be 36.1wt%, similar to the result derived from TGA (37wt%).

According to the reviewer, we tested the weight loss for TpPa-SCOF-An through TGA for comparison with SiO₂ and SiO₂@TpPa-An.

The revision has been made in the main text, *Page 15, Line 11-14*, “The outer shell accounted for ~37wt% of the total and remained the characteristic composition and structure of TpPa-COF-An verified by elemental analysis, TGA, PXRD, FT IR, and N₂ sorption (**Supplementary Fig. 24 and Supplementary Table 3**).”

Supplementary Table 3. The COF loading content of SiO₂@TpPa-An microspheres derived from ICP-AES analysis.

Sample	Si content (wt%)	COF content (wt%)
SiO ₂	12.73	/
SiO ₂ @TpPa-An	8.14	36.1

Supplementary Fig. 24 (e) TGA curves of SiO₂, SiO₂@TpPa-An, and TpPa-SCOF-An. After 700°C, all organic components were decomposed completely. Thus, compared to SiO₂ and TpPa-SCOF-An, the weight percent of TpPa-An shell in SiO₂@TpPa-An was calculated to be 37wt%.

(7) For the supplementary table 2, more references should be covered, especially more recent papers reporting high HER.

Response: Thanks for your suggestion. We have added more references reporting high

HER in *Supplementary Table 2*.

The added references are as follows.

“[S30] Li, C., Liu, J., Li, H. et al. Covalent organic frameworks with high quantum efficiency in sacrificial photocatalytic hydrogen evolution. *Nat. Commun.* 13, 2357, doi:10.1038/s41467-022-30035-x (2022).

[S31] Li, W., et al. Tuning Molecular Chromophores of Isoreticular Covalent Organic Frameworks for Visible Light-Induced Hydrogen Generation. *Adv. Funct. Mater.*, 32, 2207394, doi:10.1002/adfm.202207394 (2022)

[S32] Zhong, Y., et al. Oligo(phenylenevinylene)-Based Covalent Organic Frameworks with Kagome Lattice for Boosting Photocatalytic Hydrogen Evolution. *Adv. Mater.*, 36, 2308251. doi:10.1002/adma.202308251 (2024)”

The above references are added in *Supplementary Table 2*.

Catalyst	Co-catalyst	Sacrificial agent	HER ($\mu\text{mol g}^{-1}\text{h}^{-1}$)	AQE (%)	Ref.
CYANO-CON	Pt	Ascorbic acid	134,200	82.6 (450 nm)	[S30]
USTB-10	Pt	Ascorbic acid	21,800	0.68 (420 nm)	[S31]
COF-954	Pt	Ascorbic acid	137,230	5.00 (550 nm)	[S32]

(8) For the second step of preparation, 3 days was used. By varying the reaction time, will the crystallinity be finely tuned? The time-dependent experiments might be studied for the effect of crystallinity on the HER performance.

Response: Thanks for your suggestion. Following the preparation of the amorphous TpPa-polymer, we varied the solvothermal time (0, 12, 24, 48, and 72 hours) for the controlled crystallization via the structural rearrangement. The crystallinity of TpPa-SCOF-An can be finely tuned; longer reaction time leads to higher crystallinity (**Supplementary Fig. 19a**). Next, we investigated the photocatalytic performances of those samples (**Supplementary Fig. 19b**). Upon exposure to visible irradiation, they exhibited a linear increase in hydrogen evolution, with the HERs calculated to be 81, 296, 472, 541, and 630 $\mu\text{mol h}^{-1}$ for the samples synthesized for 0, 12, 24, 48, and 72 hours under solvothermal conditions, respectively. These results demonstrate the strong correlation between COF crystallinity and the HER performance, and imply the beneficial role of the ordered surface structure in enhancing the photocatalytic activity.

The related discussion has been added in the main text, *Page 14, Line 13-15*, “The surface crystallinity of TpPa-SCOF-An can be finely tuned by varying the solvothermal reaction time, resulting in adjustable HER performance (**Supplementary Fig. 20**). This indicates that precise control over surface ordered structure can directly influence the photocatalytic activity.” The related figure has been added in the *Supplementary Fig. 20*, as shown below.

Supplementary Fig. 20 (a) PXRD patterns and (b) time-dependent hydrogen evolution curves for TpPa-SCOF-An synthesized by varying solvothermal reaction times.

(9) The page number was missing for ref. 2.

Response: Thanks for pointing out the mistake. We have added the missing page number for ref. 2.

The related reference has been revised. “2 Low, J., Yu, J., Jaroniec, M., Wageh, S. & Al-Ghamdi, A. A. Heterojunction Photocatalysts. *Adv. Mater.* **29**, 1601694, doi:10.1002/adma.201601694 (2017).”

Reviewer #3 (Remarks to the Author):

This manuscript reported a zone crystallization strategy for COFs to enhance the surface ordering of spherical COFs through the regulator-induced amorphous-to-crystallinity transformation. The visible photocatalytic hydrogen evolution rate can reach an exceptionally high value of 126 mmol g⁻¹ h⁻¹. However, the connection between the structure of the materials synthesized in the manuscript and the synthesis strategy remains unclear. It is not evident from the paper whether materials with similar structures can be consistently synthesized under identical conditions, as there is a lack of supporting replication data. Moreover, the manuscript contains many problems as I mentioned below. The revised version can be published in Nature Communications.

1. The author should clearly specify what the regulator is in the abstract.

Response: Thanks for your suggestion. In our work, the regulator refers to the monofunctional monomer, which participates in the reversible aldimine reaction of monomers to help perfect the crystallization. We have specified the regulator's definition in the abstract.

The related content has been revised in the main text, *Page 2, line 7-11*, “As evidenced by dynamic simulations, the regulators (monofunctional monomers) initially linked on the surface of amorphous spheres play a kernel role of intensifying the surface dynamic reversibility, leading to the inner-to-outer increase in crystallinity as opposed to the bottom-up COF growth.”

2. How is the shape of COF controlled to be spherical? Has only this type of COF (TpPa-SCOF) been studied? Has there been any research on the effects of the strategies proposed in the article on the surface crystallinity and catalytic activity of other types of COFs? Is this strategy universally applicable?

Response: Thanks for your issues. In our early work (*Refs. 33,34*), we systematically investigated the amorphous-to-crystalline transformation methodology for customizing the COF morphology and size. The first-step reaction conditions determined the product shape. Dropwise addition of amines into the ethanol solution of aldehydes results in the formation of long fibers in the precipitation polymerization, while quickly mixing the two monomers produces uniform microspheres. Then, the already formed shape and size can remain in the following solvothermal conditions, endowing the final COF with specific morphology and controlled size. Besides TpPa-SCOF, other spherical COFs have been prepared with the same method in this work.

Unlike our previous reports, our current work is a novel exploration that aims to enhance regional crystallinity via regulator-mediated amorphous-to-crystalline transformation. To the best of our knowledge, this is the first report on the effect of this proposed strategy on the surface crystallinity and catalytic activity of COFs.

To validate the universality of zone crystallization strategy, we adopted 2,5-diaminopyridine (Py) and benzidine (BD) as another two linkers to fabricate the spherical COFs, *i.e.* TpPy-SCOF-An and TpBD-SCOF-An, using the An-mediated amorphous-to-crystalline transformation strategy. The newly synthesized products exhibit enhanced crystallinity, uniformly spherical morphology, and controlled sizes. The visible photocatalytic tests unequivocally confirm the success of our approach, with significantly higher HERs of TpPy-SCOF-An ($507 \mu\text{mol h}^{-1}$) and TpBD-SCOF-An ($192 \mu\text{mol h}^{-1}$) than their corresponding control samples. These results not only validate this zone crystallization strategy but also inspire future research in this field.

The related discussion has been added in the main text, *Page 14, Line 17-22 and Page 15, Line 1-4*, “To validate the universality of our strategy, we employed two representative amines, 2,5-diaminopyridine (Py) and benzidine (BD), to fabricate spherical COFs with enhanced surface crystallinity for high-performance photocatalysts (**Supplementary Figs. 21 and 22**). The photocatalytic HERs of TpPy-SCOF-An and TpBD-SCOF-An achieved the remarkable values of $507 \mu\text{mol h}^{-1}$ and $192 \mu\text{mol h}^{-1}$, respectively. These were six times and double higher than their counterparts synthesized via the direct solvothermal method. This substantial improvement in photocatalytic performance underscores the effectiveness and broad applicability of our zone crystallization strategy for enhancing the photocatalytic activity of COFs, making it a potent approach for developing organic photocatalysts.”

The synthetic method has been added in *Supplementary Information, 3. Methods*, “3.4 Synthesis of TpPy-COF and TpPy-SCOF” and “3.5 Synthesis of TpBD-COF and TpBD-SCOF”.

The related figures have been added in *Supplementary Fig. 21* and *Supplementary Fig. 22* as shown below.

Supplementary Fig. 21 TEM images for (a) TpPy-COF, (b) TpPy-COF-An, (c) TpPy-SCOF, and (d) TpPy-SCOF-An. (e) Chemical structure of TpPy-COF. (f) PXRD patterns, (g) time course for photocatalytic H₂ production under visible light irradiation, and (h) average H₂ production rates for different photocatalysts. The average diameters of TpPy-SCOF and TpPy-SCOF-An were ~400 nm. The HER values were 87, 211, 297, and 507 μmol h⁻¹, respectively.

Supplementary Fig. 22 TEM images for (a) TpBD-COF, (b) TpBD-COF-An, (c) TpBD-SCOF, and (d) TpBD-SCOF-An. (e) Chemical structure of TpBD-COF. (f) PXRD patterns, (g) time course for photocatalytic H₂ production under visible light irradiation, and (h) average H₂ production rates for different photocatalysts. The average diameters of TpBD-SCOF and TpBD-SCOF-An were ~300 nm. The HER values were 98, 133, 120, and 192 μmol h⁻¹, respectively.

3. It is recommended to use HRTEM (High-Resolution Transmission Electron Microscopy) to provide further evidence of the improvement in surface crystallinity.

Response: Thanks for your suggestion. We observed the different regions of the spherical COF surface via HR TEM. TpPa-SCOF-An possesses a large ordered area near the surface, stemming from the periodic pore arrangement. In stark contrast, the crystalline domains are nearly invisible on the surface of TpPa-SCOF. The comparison reflects that the regulator-induced synthesis does benefit crystallization for the outer moiety.

The related discussion has been revised in the main text, *Page 10, Line 6-9*, “The magnified view of the TpPa-SCOF-An surface revealed the distinct ordered domains, indicative of the high outer crystallinity, which was in stark contrast to that of TpPa-SCOF without An regulation (Fig. 2e and Supplementary Fig. 5).”

The related Figures has been added in *Supplementary Fig. 5* as shown below.

Supplementary Fig. 5 HR TEM images of the surface crystalline structures for (a) TpPa-SCOF-An and (b) TpPa-SCOF. The comparison shows the improved ordered pore alignment on the surface of TpPa-SCOF-An.

4. As we can see from Supplementary Fig.8, the crystallinity of TpPa-COF is also remarkably enhanced with assistance of An regulator. Why the author claim that “the localized ordered domains within the outer COF solids is not purposefully increased” ? The author should provide more evidence.

Response: Thanks for your issue. Our claim means that TpPa-COF-An undergoes a typical crystallization process, involving nucleation and growth along the preferential crystal facet. Hence, the crystallization evolution allows for highly ordered domains to form within the inner frameworks. The outer moiety typically bears unreactive groups or defective sites. In contrast, the crystallization of the pre-formed amorphous frameworks follows the structural rearrangement mechanism. The presence of a regulator significantly influences the transamination kinetics, leading to an increase in ordered domains near the surface. This is also a solid-to-liquid interface, making the reaction dynamics intensify pronouncedly.

To insightfully understand the regulator-mediated crystallization kinetics, we simulated the growth of TpPa-COF with the assistance of aniline in the bulk phase using the bottom-up crystallization method (**Supplementary Fig. 2**), in which aniline exists at the beginning, leading to the increased reaction reversibility (with a smaller E_{bind}). The calculated crystallization quality (Q_c) of TpPa-COF-An could be gradually decreased from inner to outer layers. This is remarkably different from the amorphous-to-crystalline route, which originates from the structural rearrangement. When the spherical surface of amorphous frameworks is pre-treated with aniline, the different crystallization kinetics between the inner and outer moiety are caused.

The related discussion has been added in the main text, *Page 8, Line 5-8*, “Additionally, simulations for the bottom-up crystallization of the bulk COF with regulators revealed that the inner layers have higher crystallinity than the outer layers (**Supplementary Fig. 2**). This unequivocally reflects the difference between the two crystallization pathways in zone crystalline regulation.”

The related figures have been added in *Supplementary Fig. 2* as shown below.

Supplementary Fig. 2 (a) The snapshot of coarse-grained molecular dynamics simulation of COF crystallization via the bottom-up method with aniline in the bulk phase. The different layers of the bulk TpPa-COF-An are marked by different colors. (b) Crystallization quality (Q_c) as a function of the position of TpPa-COF-An layers at $E_{\text{bind}} = 9.0 \text{ k}_B \text{ T}$.

5. How can the authors determine if the enhancement in catalytic activity of COFs after the addition of aniline is due to increased surface crystallinity? Is it related to an increase in surface area? Which has a greater impact, crystallinity or surface area?

Response: Thanks for your issues. We assume that the increased surface areas of COFs play a key role in the water infiltration and contact with photocatalysts, so we conducted the water vapor uptake measurement to investigate the effect of porosity. TpPa-SCOF-An, TpPa-SCOF, TpPa-COF-An and TpPa-COF afford the similar water adsorption capacity of 32.5, 34.8, 30.2, and 32.1wt%, respectively. It indicates that the optimum photocatalytic activity of TpPa-SCOF-An is not due to the increase in surface areas but crystallinity.

The related discussion has been added in the main text, *Page 13, Line 5-7*, “Additionally, the water vapor uptake measurement verified that all the COFs possessed similar water absorption capability and wetting ability (**Supplementary Figs. 15, 16**).”

The related figure has been added in *Supplementary Fig. 15* as shown below.

Supplementary Fig. 15 Water adsorption isotherms (filled symbols) and desorption isotherms (open symbols) for TpPa-SCOF-An, TpPa-SCOF, TpPa-COF-An, and TpPa-COF. The water adsorption capacities were 32.5, 34.8, 30.2, and 32.1 wt%, respectively.

6. What are the Pt contents in the catalysts shown in Fig. 5? This is crucial for comparing their reaction activities.

Response: Thanks for your issue. Through the study on the photocatalytic performance, the feeding Pt precursor was optimized to be 0.7wt% and the exact deposition contents of Pt nanoparticles were identified to be 0.14wt%, 0.29wt%, 0.55wt%, and 0.64wt% for TpPa-COF-An, TpPa-SCOF, TpPa-COF-An, and TpPa-COF, respectively, using ICP spectrometer. TpPa-COF-An achieved a maximum HER value, followed by TpPa-SCOF, TpPa-COF-An, and TpPa-COF in sequence. Thus, the lowest Pt content is responsible for the highest HER, implying that an increase in co-catalyst Pt is adverse to the photocatalytic activity. We also varied the added quantity of Pt precursor to ensure that the deposited contents were 0.1-0.2wt% for all the samples. The HER change trend remained consistent. Therefore, we can confidently state that the difference of photocatalytic activity for those samples are not originate from the co-catalyst influence.

The related discussion has been added in the main text, *Page 13, Line 17-19*, “The exact loading amount of Pt nanoparticles was as low as 0.14wt%, as confirmed by the

inductively coupled plasma (ICP) spectrometer when feeding the 0.7wt% Pt precursor.” *Page 13, Line 22 and Page 14, Line 1-5*, “With the added identical Pt precursor (0.7wt%), the control samples of TpPa-SCOF, TpPa-COF-An, and TpPa-COF deposited 0.29wt%, 0.55wt%, and 0.64wt% of Pt nanoparticles, respectively, while achieving a decrease in the HER of 451, 203, and 165 $\mu\text{mol h}^{-1}$ (Figs. 3e, 3f). Although the deposited Pt contents for the control samples were adjusted to be 0.1-0.2 wt% similar to that of TpPa-SCOF-An, the corresponding HER values were reduced continuously (Supplementary Fig. 19).”

Supplementary Fig. 19 H_2 production rates for the different photocatalysts loaded with the similar Pt amount measured by ICP, which were 0.14, 0.19, 0.16, and 0.12 wt%, respectively, for TpPa-SCOF-An-Pt, TpPa-SCOF-Pt, TpPa-COF-An-Pt, and TpPa-COF-Pt. The results suggested that the observed difference in photocatalytic activity is not due to variation in the exact Pt loading amounts.

7. The effect of growing pathways on zone crystallization of COF in Scheme 1 are quite confusing. Could you use a clearer layout to better convey the differences between these two methods?

Response: Thanks for your suggestion. The main difference between the two methods lies in that the added regulator alters the crystallization kinetics process. As shown in Scheme 1, mixing regulators with monomers for the bottom-up synthetic route results in higher crystallization in the inner region of COF, while immobilizing regulators on the surface of the precursor framework for the amorphous-to-crystalline transformation route leads to higher crystallization in the outer region of COF.

The Scheme 1 has been revised as follow.

(a) Bottom-up crystallization

(b) Amorphous-to-crystalline transformation

Scheme 1. | The effect of growing pathways on zone crystallization of COF. **a** Regulator-induced bottom-up growth of COF with outer-to-inner increase in crystallinity, only able to generate localized surface photocarriers. **b** Regulator-induced amorphous-to-crystalline transformation of COF with inner-to-outer increase in crystallinity, conducive to generate delocalized surface photocarriers.

8. More relevant references about the COFs crystallinity and photocatalytic performance should be cited, e.g. DOI: 10.1038/s41586-022-04443-4; 10.1038/s41467-023-42757-7;.

Response: Thanks for your suggestion. The two references emphasis on the COF with well-organized architecture from novel synthesis procedure for achieving excellent photocatalytic performance. We have added them in the introduction, i.e., **Ref 31,32**.

The added references are as follows.

“31 Zhang, W., Chen, L., Dai, S. *et al.* Reconstructed covalent organic frameworks. *Nature* 604, 72–79, doi:10.1038/s41586-022-04443-4 (2022).

32 Zhou, W., Wang, X., Zhao, W. *et al.* Photocatalytic CO₂ reduction to syngas using metallosalen covalent organic frameworks. *Nat. Commun.* 14, 6971, doi: 10.1038/s41467-023-42757-7 (2023).”

9. The differences in the high-resolution XPS spectra between Pt 4f for TpPa-SCOF - An/Pt and TpPa-COF-An/Pt are not obvious. The statement “The findings manifest that the outer crystalline framework with less structural defects is strongly interacted with Pt by the abundant π -electrons leading to the change of vacant 5d band orbital of Pt.” lacks sufficient evidence.

Response: Thanks for your issue. As is known from the previous reports (*Ref. 49-51*), the XPS 4f Pt peak is generally shifted in the range of 0.27-0.8 eV. In our work, we observed a 0.4-eV shift of the 4f Pt signal when comparing the XPS spectra of TpPa-SCOF-An/Pt with TpPa-COF-An/Pt. This shift is attributed to the stronger interaction of Pt with TpPa-SCOF-An than TpPa-COF-An, as earlier reported. Thus, the enhanced charge transfer from COF to Pt promotes the photocatalytic activity of TpPa-SCOF-An/Pt.

The added references are as follows.

“49 Vinayan, B. P. et al. Platinum–TM (TM = Fe, Co) alloy nanoparticles dispersed nitrogen doped (reduced graphene oxide-multiwalled carbon nanotube) hybrid structure cathode electrocatalysts for high performance PEMFC applications. *Nanoscale*, **5**, 5109-5118, doi:10.1039/C3NR00585B (2013).

50 Yu, H. et al. Donor–acceptor covalent organic framework hollow microspheres with a hierarchical pore structure for visible-light-driven H₂ evolution. *J. Mater. Chem. A*, **10**, 11010-11018 doi:10.1039/D2TA01058E (2022).”

10. Did the author experiment with the maximum concentration limit of aniline, beyond which COFs cannot be formed? Please report on this.

Response: Thanks for your issue. We have supplemented the experiments for studying the effect of aniline concentrations on the COF formation during the structural transformation. When 300 *equiv.* of aniline was used, the post-reaction solution was transparent without any precipitates obtained via the amorphous-to-crystalline route (**Supplementary Fig. 7**). It is most likely that the forward reaction between aldehydes and aniline is predominated to suppress the formation of COF frameworks.

The related figure has been added in *Supplementary Fig. 7* as shown below.

Supplementary Fig. 7 Photograph of the post-reaction solution using excessive aniline (300 *equiv.*) for synthesizing TpPa-SCOF-An.

Reviewer #4 (Remarks to the Author):

In the present manuscript, Lin and co-workers have reported the effect of nanoscale crystallinity over the photocatalytic hydrogen evolution, which they have denoted as the “zone crystallization”. The crystallinity of the COF nanoparticles was tuned by using a modulator, leading to the superhigh rate of photocatalytic hydrogen evolution. After reading the manuscript carefully, I find the manuscript lacks novelty that required for publishing in Nature Communications for the following reasons:

a) Transforming kinetically controlled amorphous polymer into thermodynamically stable crystalline COFs is well documented COF chemistry. Also, the usage of modulator to tune or improve the crystallinity has been extensively reported by various research groups.

Response: Thanks for your comment. We would like to highlight the novelty and significance of our work for your ease of re-evaluation.

Firstly, the adoption of an amorphous-to-crystalline transformation strategy is based on our early work. In 2016, we explored a methodology of transforming kinetically controlled amorphous polyazomethine into thermodynamically stable keto-enamine-linked COF (*Ref. 33*). We precisely regulated the COF size and morphology, constructed the COF-based core/shell microspheres, and merged functionalities of different materials. In 2020, we varied the reaction conditions of the amorphous-to-crystalline method to modulate the COF morphology into fibers and spheres (*Ref. 34*). This time, we introduced the regulator attached to the surface of the amorphous frameworks for strengthening the surface crystallinity of the spherical COF through the identical amorphous-to-crystalline route. We pioneer the exploration of the zone-crystallization regulation method, which has not been reported anywhere yet.

Secondly, the utilization of regulators to increase COF crystallinity has been documented for the bottom-up growth of COFs (*Refs. 35,36*). Differing from those previous reports, we immobilized the regulator (aniline) onto the amorphous frameworks, which was then subject to the amorphous-to-crystalline conversion. The regulator-induced structural rearrangement was predominated on the spherical surface, leading to an increase in outer crystallinity. Without sophisticated molecular design, such the simplest COF can achieve the ultrahigh photocatalytic H₂ evolution rate (630 $\mu\text{mol h}^{-1}$), surpassing the other control COFs synthesized using the bottom-up solvothermal method or conventional amorphous-to-crystalline route. Therefore, we confer an insightful understanding of the strong connection between photoactive ordering surface and photocatalytic performance, which is the first report for COF-based photocatalysts.

Finally, we appreciate your comment and acknowledge that the differences may not have been expressed clearly enough in previous versions of the manuscript. We have

revised the manuscript, particularly the introduction and the discussion of the photocatalytic performance, to more clearly illustrate the novelty of our work.

The related discussion has been added in the main text, *Page 5, Line 4-8*, “In our early report, we demonstrated the controllability of the morphology, size and structure of COFs at the microscale via an amorphous-to-crystalline transformation.^{33,34} This pathway also holds great promise for surface crystallization engineering with regulators, which have often been incorporated for enhanced crystallinity of bulk COFs.^{35,36}”

The added references are as follow.

“33 Tan, J. *et al.* Manipulation of Amorphous-to-Crystalline Transformation: Towards the Construction of Covalent Organic Framework Hybrid Microspheres with NIR Photothermal Conversion Ability. *Angew. Chem. Int. Ed.* 55, 13979, doi: 10.1002/anie.201606155 (2016).

36 Wang, S. *et al.* Reversible Polycondensation-Termination Growth of Covalent-Organic-Framework Spheres, Fibers, and Films. *Matter* 1, 1592-1605, doi: 10.1016/j.matt.2019.08.019 (2019).”

b) Although the Scheme 1 is visually appealing and convincing, the manuscript fails to provide experimental evidence that support the hypothesis. Specifically, it is unclear how the authors distinguish experimentally between localized and delocalized surface carriers.

Response: Thanks for your insightful comment. The localized and delocalized surface carriers can be distinguished by the TAS kinetic analysis in **Fig. 4d**. Specifically, the photogenerated electron lifetime of TpPa-SCOF-An is as long as 233.96 ps, which is 164-fold longer than that of TpPa-COF-An (1.43 ps). The extended lifetime for TpPa-SCOF-An suggests a delocalized carrier nature, reducing opportunities for charge recombination and thereby enhancing photocatalytic redox reactions. Conversely, a much shorter lifetime on TpPa-COF-An indicates a predominance of localized surface carriers, which are more prone to recombination and thus impede the overall photocatalytic performance.

The related discussion has been added in the main text, *Page 17, Line 9-13*, “The extended lifetime suggests a delocalized carrier nature on the surface of TpPa-SCOF-An, reducing the opportunity for charge recombination.⁴⁸ Conversely, a much shorter lifetime on TpPa-COF-An indicates a predominance of localized surface carriers, which are more prone to be recombined and impede the overall photocatalytic performance.”

The added reference is as follows.

“48 Zhang, S. *et al.* Strong-base-assisted synthesis of a crystalline covalent triazine framework with high hydrophilicity via benzylamine monomer for photocatalytic water splitting. *Angew. Chem. Int. Ed.* 59, 6007-6014, doi: 10.1002/anie.201914424 (2020).”

c) While it is fundamentally established that a built-in electric field helps to separate the charge carriers, however, the manuscript does not clarify how crystalline COF has high built-in electric field compared to the amorphous counterpart. I am sceptical about the KPFM results, due to the lack of detailed methodology, including sample preparation and experimental conditions, moreover the measurement was conducted for a single nanoparticle. To validate the result, more than 100 COF particles should be screened.

Response: Thanks for your suggestion. We have provided a detailed methodology for the KPFM measurement, including sample preparation and experimental conditions.

We agree that only one single nanoparticle is not enough to reflect the intrinsic nature of samples. The KPFM measurement for other more nanoparticles has been conducted to collect more surface potential images. As those results are similar, the representative images are shown here (**Fig. R2**). TpPa-SCOF-An and TpPa-SCOF give the surface potentials of 31.40 and 14.69 mV, respectively, which are similar to the values given in the manuscript. However, we do not have adequate testing time to collect 100 images, and it is also hard to take many nanoparticles in one image due to the resolution limit. Therefore, we are not confined to KPFM for microscopic characteristics but explore other methods to investigate macroscopic surface potentials.

As previously reported (*Ref. 57*), the surface photovoltage (SPV) measurement is a potent tool for acquiring surface potentials, providing an average result for the entire sample. In line with this, we conducted the SPV spectra tests for the two samples. The surface voltage intensity of TpPa-SCOF-An and TpPa-SCOF was found to be 84.5 and 40.5 μV , respectively. The corresponding BEF intensity of TpPa-SCOF-An was calculated to be 1.74 times higher than that of TpPa-SCOF, which aligns well with the KPFM findings. It also proves that the microscopic surface potentials derived from KPFM in the manuscript are reliable.

The related discussion has been revised in the main text, *Page 19, Line 14-19*, “Using the Kelvin Probe Force microscopy (KPFM) to observe the discrete particles, the microscopic surface potential of TpPa-SCOF-An was measured to be 29.15 mV, which was almost twice higher than that of TpPa-SCOF (14.04 mV) (**Fig. 5e** and **Supplementary Fig. 31**). Meanwhile, we carried out the surface photovoltage test for the macroscopic surface potentials, which were well in line with the KPFM results (**Supplementary Fig. 32**).⁵⁷”

The sample preparation and experimental condition have been added in *Supplementary information, Section 3.10*, “The microscopic surface potential V_s of the single particle is measured by Kelvin Probe Force Microscopy (KPFM) in the mode of surface potential (**Supplementary Fig. 31**). Specifically, the newly synthesized SCOF sample (1 mg) were dispersed in 10 mL anhydrous ethanol and sonicated with a frequency of

40 kHz at room temperature for 30 min. 10 μL of the dispersion was placed on the surface of the silicon chip and dried at room temperature to remove the solvent for the further KPFM test. For each measurement, a known bias is applied between the sample and the tip of the needle to obtain the surface potential distribution of the sample.” And “The macroscopic surface potential V_s of the whole sample is acquired by measuring the surface photovoltage spectra (**Supplementary Fig. 32**). The powder sample was sandwiched between two ITO glass. The test system includes a Xe light source, a monochromator, a chopper, and a lock-in amplifier.”

The related figures have been added in *Supplementary Figs. 32* as shown below.

Supplementary Fig. 32 (a) Surface photovoltage for TpPa-SCOF-An and TpPa-SCOF and (b) the comparison of surface potential derived from SPV and KPFM, respectively.

The related reference is added as below.

“57 Jing, J. et al. Construction of Interfacial Electric Field via Dual-Porphyrin Heterostructure Boosting Photocatalytic Hydrogen Evolution. *Adv. Mater.* **34**, 2106807, doi: 10.1002/adma.202106807 (2022).”

Fig. R2 (a) The height image and (c) the corresponding surface potential image of TpPa-SCOF. (b) The height image and (d) the corresponding surface potential image of TpPa-SCOF-An.

d) The reported rate of photocatalytic hydrogen evolution is exceptionally high, which need to be verified. While, I do not doubt the author’s results, however previously, it was noticed several instances that the photocatalytic activity varied significantly under different experimental conditions.

Response: Thanks for your suggestion. To demonstrate the reliability of photocatalytic hydrogen evolution performance, we conducted photocatalysis experiments using three different batches of the COF materials under the same conditions. The results revealed that all three batches of the TpPa-SCOF-An samples exhibited the similar photocatalytic performance. Meanwhile, we provided the precise experimental conditions including irradiation intensity and light source spectra to ensure the test repeatability.

The related discussion has been revised in the main text, *Page 13, Line 8-10*, “The photocatalytic hydrogen evolution experiments were performed under visible irradiation ($\lambda > 420$ nm) using ascorbic acid as a sacrificial electron donor and Pt as a cocatalyst (**Supplementary Fig. 17**).” *Page 13, Line 13-15*, “Such an ultrahigh HER could be reproduced under identical photocatalytic conditions for TpPa-SCOF-An synthesized from three different batches (**Supplementary Fig. 18**).”

The related figures have been added in *Supplementary Figs. 17 and 18*, as shown below.

Supplementary Fig. 17 (a) Photograph of the device used in the photocatalytic H₂ evolution tests. (b) Scheme of the light irradiation condition. (c) The irradiation intensity measured for the position illustrated in Scheme b. (d) Light source spectra for the 300 W Xe lamp equipped with a cut-off filter (>420 nm).

Supplementary Fig. 18 Time course for photocatalytic H₂ production using TpPa-SCOF-An from three different batches.

REVIEWER COMMENTS

=====
Reviewer #1 (Remarks to the Author):

I commend the authors for their significant effort in addressing the feedback I provided. The manuscript has been substantially improved as a result. Based on these improvements, I support the acceptance of this work for publication

Response: We greatly appreciate the positive recommendation from the reviewer.

=====
Reviewer #2 (Remarks to the Author):

I think that the authors have well addressed the comments made by the reviewers in the revised version of the manuscript. Therefore, I have no further questions. It should be accepted.

Response: We sincerely appreciate the reviewer's favorable recommendation.

=====
Reviewer #3 (Remarks to the Author):

I have carefully reviewed the revised version of the manuscript and Point-by-point Response to the Comments. Basically met the revision requirements and agreed to be published.

Response: We greatly appreciate the support and positive feedback from the reviewer.

Reviewer #4 (Remarks to the Author):

I have gone through the revised manuscript and response letter. However, still I am not convinced by the author's justifications, because of the following reasons.

a) the concept of zone crystallization is not validated as shown in scheme 1, and it does not match with TEM image in SI fig 5. It's still more of an amorphous to crystalline transformation.

Response: Thanks for the reviewer's issue. The HR TEM images in **Supplementary Fig. 5** shows the two particles of TpPa-SCOF-An and TpPa-SCOF, which were prepared both via the two-step route of amorphous-to-crystalline transformation. The difference is that the synthesis of TpPa-SCOF-An incorporated the regulator aniline during the structural transformation for enhancing the surface crystallinity. Thus, the more remarkable lattice regions on the surface of microsphere were observed for TpPa-SCOF-An than that of TpPa-SCOF synthesized without use of aniline.

As previously reported (A. I. Cooper, et al. J. Am. Chem. Soc. 2020, 142, 25, 11131), the high crystallinity of 2D COFs plays a crucial role in maintaining the linkage stability in acid/base environments, thanks to the densely layered stacking. The amorphous-to-crystalline transformation is accompanied by the formation of defective structures derived from unreactive sites or error connections. This suggests that the size and distribution of crystalline domains are closely linked to the structural stability of COF microspheres. With this in mind, we present additional evidence for the surface-enhanced crystallinity of TpPa-SCOF-An. The sample was immersed in a 3 M NaOH aqueous solution for COF digestion and then observed for morphological changes, which reflect the differences in crystalline regions.

After alkali erosion, TpPa-SCOF-An reveals remarkable internal cavities or slits, as displayed in the TEM image (**Supplementary Fig. 9a**). In contrast, TpPa-SCOF appears solid inside and quite rough on the surface, resembling the assembling of numerous small grains (**Supplementary Fig. 9b**). This contrast highlights the role of aniline in enhancing the crystallinity on the surface of TpPa-SCOF-An, thereby increasing its alkali resistance and maintaining the stability of the peripheral moiety to some extent. The observed internal cavities provide evidence that the internal structure is readily etched primarily due to the more defective sites. In contrast, TpPa-SCOF synthesized without the use of aniline has fewer crystalline domains on the surface, making it more susceptible to alkali corrosion. These findings underline the role of aniline in controlling the amorphous-to-crystalline transformation for the zone crystallization of COF microspheres.

The related discussion has been revised in the main text, **Page 11, Line 7-13**, "On the other hand, we conducted an experiment on strong base erosion as the morphological change caused by harsh chemical environments can reflect the extent of COF

crystallinity. TpPa-SCOF-An generated internal cavity or slits, while TpPa-SCOF appeared solid insides and rough on the surface, resembling the accumulation of small grains (Supplementary Fig. 9). This contrast is probably due to the surface ordering increase of TpPa-SCOF-An, leading to the sluggish erosion for the peripheral moiety.”

The related figure has been added in the *Supplementary Fig. 9* as shown below.

Supplementary Fig. 9 TEM images of (a) TpPa-SCOF-An and (b) TpPa-SCOF exposed to 3 M NaOH aqueous solution for 24 hours. The recovered solids were further rinsed with water and THF several times. One can see that TpPa-SCOF-An appears internal cavities or slits and smooth surface, while TpPa-SCOF is changed to be accumulation of small grains with solid insides.

b) Regarding delocalized surface carriers, I do not agree with the authors because the long lifetime of the photogenerated electron can be attributed to the trapped electrons. Also, if I consider the kinetic fits, long lifetime component has very little population.

Response: We agree with the reviewer that the results in ultrafast TA measurements could not definitively support our conclusions. Indeed, kinetics and dynamics occurring in the ultrafast time scale mainly reflect photogenerated exciton behaviors and do not correlate well with HER performance. Therefore, we have further conducted experiments with nanosecond TA spectroscopy where long lived signals could report

the kinetics and dynamics of charge separated states. Full TA spectra and characteristic single wavelength kinetics were obtained for TpPa-SCOF-An and TpPa-COF-An, both with and without Pt cocatalysts loading. As the additional data shown in **Fig. 4** and **Supplementary Fig. 31**, TpPa-SCOF-An/Pt displayed nearly three orders of magnitude longer lifetime (98.5 μ s) than TpPa-SCOF-An, which is in stark contrast to the TpPa-COF-An counterpart. The nanosecond TA experiments provide direct spectroscopic evidence that the enhanced surface crystallinity reduces the interfacial electron transfer barrier between the COF's peripheral moiety and Pt cocatalysts for efficient electron trapping to Pt as well as retarded charge recombination for enhanced catalytic H₂ turnovers. Hence, the microsecond long-lived transients strongly support the presence of delocalized carriers on TpPa-SCOF-An.

The relevant discussion has been added in the main text, *Page 17, Line 8-22 and Page 18, Line 1-13*, “The nanosecond transient absorption (TA) spectroscopy was further employed to directly examine the kinetics and dynamics of photogenerated charge carriers in TpPa-SCOF-An and TpPa-COF-An, both with and without Pt cocatalysts loading. The full TA spectra of TpPa-SCOF-An and TpPa-COF-An both show similar broad excited state absorption (ESA) features, spanning from 350 nm to 800 nm, with decay lifetimes ranging from tens of nanoseconds to around one hundred nanoseconds (**Supplementary Fig. 31a and b**). The broad ESA band decaying on this time scale is attributed to singlet-singlet annihilation of diffusional excitons.⁴⁷ Closer examination of the two full TA spectra reveals a slight downward curvature in the 550-700 nm range for TpPa-SCOF-An relative to TpPa-COF-An, which may arise from the emergence of free charge carriers.

Upon loading with Pt cocatalysts, a prominent ground state bleach (GSB) signal emerges between 550-700 nm in the full TA spectra of TpPa-SCOF-An/Pt, with a dramatically extended lifetime of 98.5 μ s, as derived from multiexponential kinetic decay fitting at 580 nm (**Supplementary Fig. 31c, Supplementary Table 4, and Fig. 4b**). This substantial increase in lifetime, by nearly three orders of magnitude, indicates efficient charge separation from exciton singlet-singlet annihilation in TpPa-SCOF-An/Pt as well as free electron trapping at Pt catalytic sites for subsequent H₂ evolution. In contrast, the full TA spectra for Pt cocatalysts loaded TpPa-COF-An do not display GSB bands and lack similar enhancements in lifetime, suggesting inefficient charge separation in TpPa-COF-An (**Supplementary Fig. 31b, Supplementary Fig. 31d, and Fig. 4c**). The TA experiments provide direct spectroscopic evidence that the enhanced surface crystallinity in TpPa-SCOF-An reduces the interfacial electron transfer barrier between the COF's peripheral moieties and Pt cocatalysts, significantly boosting photocatalytic H₂ evolution. Therefore, the effect of An on regulating surface crystallization is more prominent in the amorphous-to-crystalline transformation than the bottom-up route, leading to the remarkable increase in surface ordering for long-lived photogenerated active states and robust photocatalytic performance.”

The related figures have been added in the *Fig. 4* and *Supplementary Fig. 31* as shown

below.

Fig. 4 | Photophysical measurements. **b** Normalized transient kinetic decays monitored at 580 nm after pulsed 532 nm excitation for TpPa-SCOF-An and **c** TpPa-COF-An, in the presence and absence of the loading of Pt cocatalyst. Bleach signals were normalized to 1 for clear comparison. The overlaid curves in orange and green are kinetic fits to the multi-exponential kinetic model.

Supplementary Fig. 31 TA spectra of (a) TpPa-SCOF-An, (b) TpPa-COF-An, (c) TpPa-SCOF-An/Pt and (d) TpPa-COF-An/Pt suspension in aqueous solution under inert atmosphere at different time delay after pulsed laser excitation ($\lambda_{ex} = 532$ nm, 5 mJ/pulse).

The related table has been added in the *Supplementary Table 3*, as shown below

Supplementary Table 4. Fitting Parameters Based on the Multi-exponential Function.

Sample	A_1	t_1 (s)	A_2	t_2 (s)	A_3	t_3 (s)	\bar{t} (s) ^a
TpPa-SCOF- An/Pt	0.783	4.82E-07	0.330	5.66E-06	0.0852	1.19E-04	9.85E-05
TpPa-SCOF- An	74.9	2.36E-08	-	-	-	-	-
TpPa-COF- An/Pt	0.991	7.81E-08	-	-	-	-	-
TpPa-COF- An	2.64	1.06E-07	-	-	-	-	-

^a The average lifetime is calculated based on the following equation,

$$\bar{t} = \frac{A_1 t_1^2 + A_2 t_2^2 + A_3 t_3^2}{A_1 t_1 + A_2 t_2 + A_3 t_3}.$$

The related reference is shown in below,

47 Jakowetz, A. C. et al. Excited-State Dynamics in Fully Conjugated 2D Covalent Organic Frameworks. *J. Am. Chem. Soc.*, **141**, 11565–11571, doi: doi.org/10.1021/jacs.9b03956 (2019).

Overall, the claims are too ambitious compared to the experimental evidence.

Response: With more experimental evidence including the alkali erosion experiments and the additional nanosecond transient absorption data that directly link to the photocatalytic HER performance, we hope that our conclusion is strengthened on the point of the increased surface crystallinity benefiting photocatalysis.

REVIEWER COMMENTS

Reviewer #4 (Remarks to the Author):

Although I do not completely agree with the claims of the present manuscript, however the authors have done a large number of supporting experiments to indirectly prove their claims. Authors are requested to address the following comments.

a) Etching with NaOH cannot selectively etch the amorphous region because principally both crystalline and amorphous domain can have imine and enamine linkage. Also, authors did not perform any analytical experiments after the etching with NaOH, such as PXRD, IR and solid state CP MAS NMR, which will provide detail information about the structure.

Response: Thanks for the reviewer's issue.

We apologize for any confusion caused by our previous response regarding the etching of materials with NaOH. To clarify, both crystalline and amorphous regions can be etched using NaOH aqueous solutions. However, we would like to emphasize that the reaction kinetics may differ between these two types of regions. As noted in earlier research by A. I. Cooper et al. (J. Am. Chem. Soc. 2020, 142, 25, 11131), the ordered layered structure of the crystalline domains in 2D COFs plays a significant role in maintaining linkage stability in acidic/basic environments. This suggests that crystalline regions may exhibit slower etching kinetics compared to amorphous regions.

Indeed, our additional data demonstrate that alkali etching results in a clear distinction in the spherical morphology observed in the TEM images (**Supplementary Fig. 9**). We attribute this contrast to the fact that the ordered layered structures in the crystalline regions provide greater chemical stability for the linkages than those found in the amorphous regions, leading to more rapid degradation in the latter.

In response to the reviewer's comments, we performed measurements of PXRD, FT IR, and solid-state CP/MAS ^{13}C NMR to compare structural and compositional changes before and after alkali etching. As anticipated, the remaining components of TpPa-SCOF and TpPa-SCOF-An exhibit identical chemical compositions. Their PXRD patterns indicate that crystallinity is significantly weakened after etching (**Supplementary Fig. 10**). Therefore, apart from observable changes in appearance, we cannot provide further details distinguishing between TpPa-SCOF and TpPa-SCOF-An.

We have revised it for ease of understanding the content in the main text, *Page 11, Line 7-17*, "On the other hand, when COF microspheres are etched using an aqueous solution of NaOH, the ordered layered structures in the crystalline regions may slow the etching process compared to the

amorphous regions, resulting in differences in their appearance. Following alkali post-treatment, TpPa-SCOF-An exhibited internal cavities or slits, while TpPa-SCOF appeared solid on the inside and rough on the surface, resembling a collection of small grains (see Supplementary Fig. 9). This contrast likely arises because the increased surface ordering of TpPa-SCOF-An leads to slower etching of the outer components. Although the observed morphologies differ, the remaining components of both samples have identical compositions, with a significant reduction in crystallinity noted after etching (see Supplementary Fig. 10).”

The related figure has been added in the **Supplementary Fig. 10** as shown below.

Supplementary Fig. 10 (a) PXRD patterns, (b) FT IR spectra, and (c) solid-state CP/MAS ¹³C NMR spectra of TpPa-SCOF-An and TpPa-SCOF before and after being exposed to 3M NaOH aqueous solution for 24 h.

b) Delocalized charge carriers should reach the Pt centers quickly and leading to fast kinetics however here we observed different trends in TAS analysis.

Response: Thanks a lot for the reviewer’s comment.

Yes, we fully agree with the reviewer that “delocalized charge carriers should reach the Pt centers quickly and leading to fast kinetics”. The observed different trends in the ns-TAS analysis actually

represented charge recombination of already formed free carriers and trapped carriers at Pt catalytic sites. The longer lifetime for charge recombination is indeed beneficial for subsequent H₂ evolution as catalysis typically occurs on much slower time scales. We are sorry for our misinterpretation in the previously revised manuscript which probably caused the confusion and thanks again for pointing out this issue. The interpretation now is sound and in line with other characterizations and conclusions.

The corresponding paragraph is now revised as, *Page 18, Line 5-10*, “The prominent GSB signal indicates efficient charge separation from exciton singlet-singlet annihilation in TpPa-SCOF-An/Pt as well as free electron trapping at Pt catalytic sites. The nearly 100 μs long lifetime process is attributed to charge recombination of free carriers and trapped carriers at Pt catalytic sites. This substantial increase in lifetime, by nearly three orders of magnitude, is beneficial for subsequent H₂ evolution as catalysis typically occurs on much slower time scales.”